# Genome-wide association analysis provides insights into the molecular etiology of dilated cardiomyopathy

Dilated cardiomyopathy (DCM) is a leading cause of heart failure and cardiac transplantation. We report a genome-wide association study and multi-trait analysis of DCM (14,256 cases) and three left ventricular traits (36,203 UK Biobank participants). We identified 80 genomic risk loci and prioritized 62 putative effector genes, including several with rare variant DCM associations (*MAP3K7*, *NEDD4L* and *SSPN*). Using single-nucleus transcriptomics, we identify cellular states, biological pathways, and intracellular communications that drive pathogenesis. We demonstrate that polygenic scores predict DCM in the general population and modify penetrance in carriers of rare DCM variants. Our findings may inform the design of genetic testing strategies that incorporate polygenic background. They also provide insights into the molecular etiology of DCM that may facilitate the development of targeted therapeutics.

Dilated cardiomyopathy (DCM) describes a spectrum of heart muscle diseases that are characterized by impaired left ventricular (LV) myocardial contractility and dilatation, in the absence of coronary artery disease (CAD) or abnormal loading conditions[1,2]. DCM affects approximately one in 250 individuals and is among the primary etiologies of heart failure, as well as the leading cause of cardiac transplantation[3]. Pathogenic variants in relevant genes can cause DCM via monogenic disease mechanisms; however, recent evidence suggests important direct and indirect effects of polygenic background on DCM risk[4]. Characterization of the complex genetic architecture underlying DCM provides opportunities for improved clinical genetic testing and the discovery of pathways and genes to inform therapeutic development.

## Results

### Genome-wide association study and multitrait analysis of dilated cardiomyopathy identifies novel genomic risk loci

We performed a meta-analysis of case–control genome-wide association studies (GWASs) comprising 14,256 DCM cases and 1,199,156 controls, from 16 studies participating in the Heart Failure Molecular Epidemiology for Therapeutic Targets (HERMES) Consortium[5] (Fig. 1, Extended Data Fig. 1, Supplementary Tables 1 and 2, and

Supplementary Information 1). Patients who meet guideline definitions of DCM may not carry the disease label, leading to incomplete ascertainment of cases[6]. To improve DCM ascertainment in large research cohorts and health record-based biobanks, we developed a phenotyping algorithm without a requirement for data on LV chamber dimensions (Supplementary Information 2), which are frequently not available in studies. Of the 16 studies, six included cases recruited from specialist clinical cohorts or unequivocal DCM diagnostic codes (DCM_Narrow: 6,001 cases (76.2% recruited from specialist clinical cohorts) and 449,382 controls), whereas 11 ascertained cases based on an inclusive definition of LV systolic dysfunction in the absence of secondary causes, without specific requirements for ventricular dilatation (DCM_Broad: 9,299 cases and 1,157,145 controls). We found complete genetic correlation between DCM_Narrow and DCM_Broad ($r_g$ = 1.00), highlighting the shared genetic architecture between these phenotype definitions, and all studies were therefore combined for meta-analysis (DCM GWAS).

Among 9,656,392 common variants (minor allele frequency (MAF) > 0.01) included in the meta-analysis, we identified 27 independent variants at 26 genomic loci passing genome-wide significance ($P < 5 \times 10^{-8}$) (Fig. 2, Extended Data Fig. 2 and Supplementary Table 3). Eighteen of the 26 loci were associations that had not been previously

✉e-mail: j.ware@imperial.ac.uk; t.lumbers@ucl.ac.uk

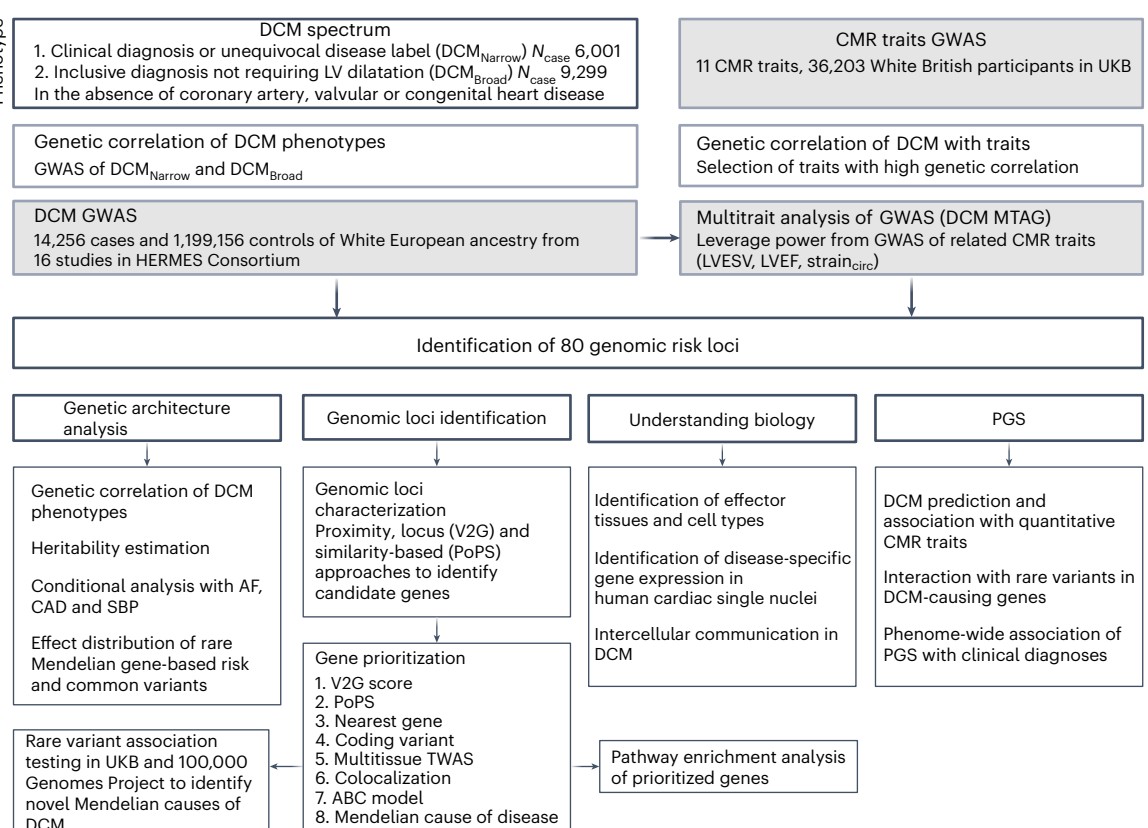

**Fig. 1 | Study overview of European ancestry DCM GWAS performed in 14,256 cases and 1,199,156 controls from 16 studies.** Cases were defined as having a clinical diagnosis or unequivocal disease label for DCM (DCM_Narrow) or a more inclusive definition of LV systolic dysfunction, with or without LV dilatation (DCM_Broad), in the absence of CAD, severe valvular heart disease or congenital heart disease. Genetic correlation was performed to identify traits suitable for inclusion in meta-analysis and multitrait analysis of GWAS (MTAG). The MTAG analysis combined DCM GWAS with GWAS of genetically correlated quantitative cardiac magnetic resonance (CMR) imaging-derived traits (DCM MTAG). Downstream analyses included elucidating the genetic architecture of DCM, genomic risk loci annotation and prioritization of candidate genes, integration with single-cell transcriptomics to identify perturbations of candidate gene expression, and generation and evaluation of polygenic risk scores (PGS) for DCM. LVESV, LV end systolic volume; LVEF, LV ejection fraction; strain_circ, global LV circumferential strain. Figure created with BioRender.com.

reported for DCM (Supplementary Tables 3 and 4). An additional 36 variants at 36 loci met the criterion of a 1% false discovery rate (FDR) (equivalent to $P < 2.2 \times 10^{-6}$).

Next, we compared the effect estimates from DCM GWAS against the subset of six studies with cases carrying a clinical diagnosis (DCM_Narrow GWAS, Extended Data Fig. 3). All 62 DCM GWAS loci identified using the 1% FDR threshold had directionally concordant effects in DCM_Narrow GWAS. Of these, ten loci reached the genome-wide significance threshold ($P < 10^{-8}$) with most having a larger effect size in DCM_Narrow GWAS (Supplementary Table 3 and Extended Data Fig. 3). Using linkage disequilibrium (LD)-adjusted kinships (LDAK) with summary statistics from GWAS[7], we estimated the heritability explained by common single-nucleotide polymorphism (SNPs; $h^2_{SNP}$) on the liability scale as 20% (2.1% s.d.) for DCM_Narrow GWAS and 11% (1% s.d.) for DCM GWAS.

To explore shared genetic etiology with quantitative LV traits and to evaluate the potential of combining traits through multitrait analysis of GWAS (MTAG), we estimated the pairwise genetic correlation ($r_g$) between DCM and ten cardiac magnetic resonance imaging-derived (CMR) traits from 36,203 participants in the UK Biobank (UKB), using bivariate LD score regression[8,9]. Three LV traits were highly correlated with DCM: end-systolic volume (LVESV), $r_g = 0.73$; global circumferential strain, $r_g = 0.71$; and ejection fraction (LVEF), $r_g = -0.70$ (Supplementary Table 5). These traits were included in a DCM-anchored MTAG (DCM MTAG), allowing for a joint analysis to increase statistical power[10].

Fifty-eight sentinel variants at 54 loci were identified at $P < 5 \times 10^{-8}$ by DCM MTAG, including 18 loci not identified in our GWAS at FDR < 1%. Twenty-eight of the 54 loci were associations not previously reported for DCM or any of the three LV traits included in the MTAG (Supplementary Tables 3 and 4).

A total of 59 genomic risk loci reached genome-wide significance in GWAS or GWAS_MTAG, 31 of which had not been previously reported to be associated with DCM or related cardiac traits (Supplementary Tables 3 and 4). Among loci identified in the DCM GWAS, 25 FDR-significant loci were not significant in DCM MTAG; however, all uniquely significant loci (DCM GWAS and DCM MTAG) had directionally concordant effects (Extended Data Fig. 3). For subsequent locus- and gene-based analyses we investigated a discovery set of 80 genomic loci, identified through either DCM GWAS (FDR < 1%) or DCM MTAG ($P < 5 \times 10^{-8}$), applying a range of orthogonal approaches to prioritize potential effector genes.

Using functionally informed fine-mapping, we identified 100 credible sets of likely causal variants at 63 of 80 loci. The credible sets consisted of 1,392 variants (60.6% intronic, 25.4% intergenic and 4.8% exonic). Among these, 83 variants identified at 43 loci had a posterior inclusion probability (PIP) > 0.5 (Extended Data Fig. 4 and Supplementary Table 6). Several fine-mapped coding variants were found within known DCM genes (*FLNC*, *BAG3* and *TTN*) and genes with plausible effects on cardiac function (*NEXN* and *MYBPC3*), including deleterious missense variants (combined annotation-dependent depletion Phred score >15) in *TTN*, *BAG3* and *MYBPC3*.

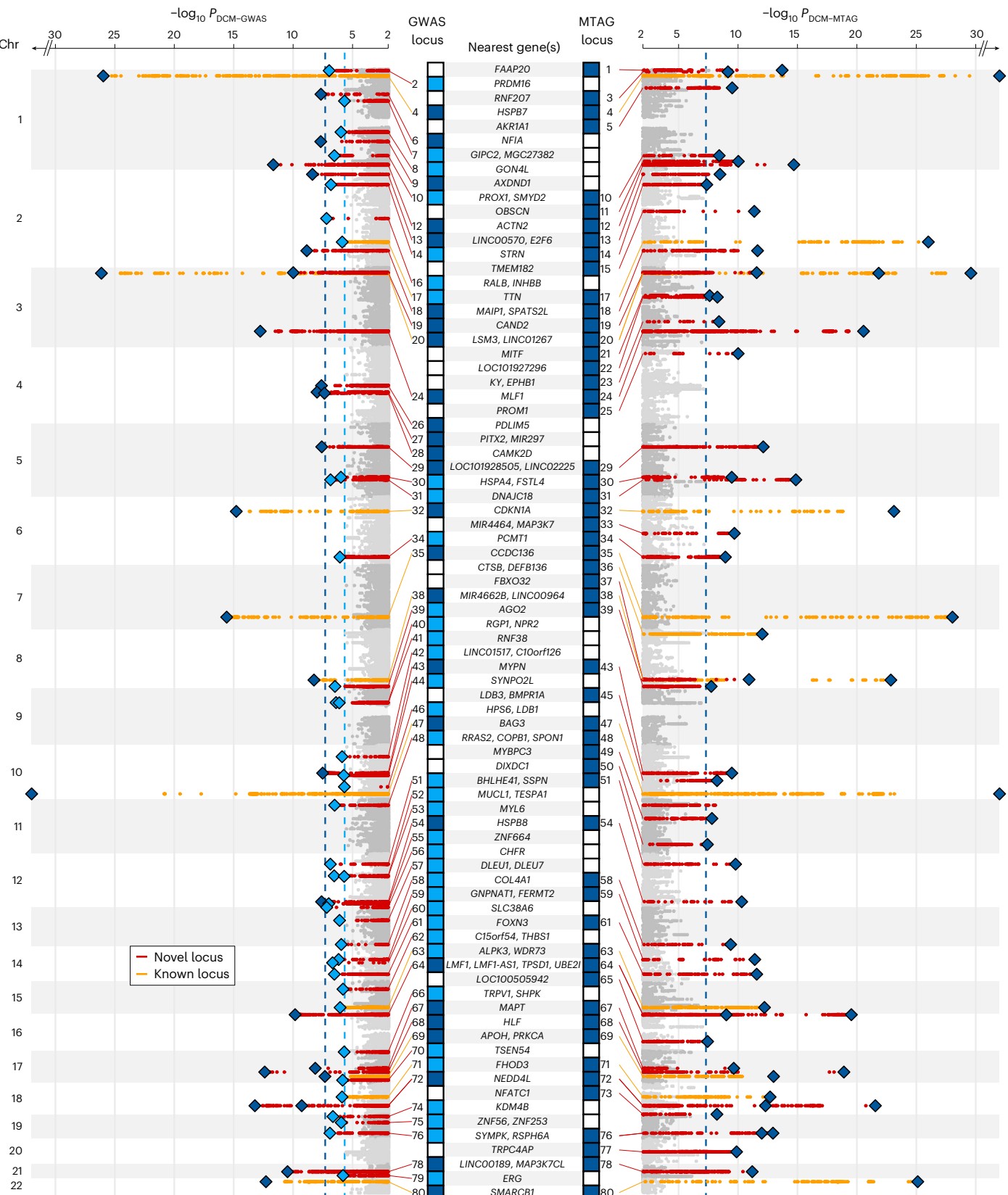

**Fig. 2 | Manhattan plot of DCM GWAS and DCM MTAG identifying novel (red) and previously reported (orange) genomic loci associated with DCM.** Loci reaching genome-wide ($P < 5 \times 10^{-8}$, dashed blue line) in DCM GWAS and DCM MTAG, and FDR ($\alpha_{FDR} < 1\%$, dashed light blue line) in DCM GWAS are highlighted.

Loci are annotated with the nearest protein-coding gene(s) of all conditionally independent variants within the locus and ordered in ascending genomic location. $P$ values were two-sided and based on an inverse-variance weighted fixed-effects model and not adjusted for multiple testing.

## Effector gene prioritization and pathway enrichment analysis identify molecular mechanisms

To prioritize effector genes for DCM, we assessed functional evidence for 1,970 protein-coding genes situated within or overlapping the identified genomic risk loci (Fig. 3a and Supplementary Table 7). First, using a combination of nearest gene, locus-based (variant-to-gene (V2G)) and similarity-based (polygenic priority score (PoPS)) methods, we identified 380 candidate genes for further prioritization (median 5 per locus; interquartile range 4–6). Second, by incorporating additional evidence from five complementary methods—coding variants, colocalization with expression quantitative trait loci (eQTL), transcriptome-wide association studies (TWAS), activity-by-contact (ABC) model, and established Mendelian cardiomyopathy- or muscle-disease-causing genes—along with results from the three initial methods, we identified a single prioritized gene at 62 of 80 loci (Fig. 3b, Extended Data Fig. 5 and Supplementary Table 8). The highest prioritization scores were for *MYPN* (prioritized by seven of the maximum of eight predictors), followed by *HSPB8* and *ALPK3* (six predictors), and *ACTN2*, *SPATS2L* and *BAG3* (five predictors). Highlighting the robustness of this framework, all ClinGen genes with definitive evidence for Mendelian cardiomyopathy, except *LMNA*, were prioritized at their respective loci. Genes associated with Mendelian forms of hypertrophic cardiomyopathy (HCM) (*MYBPC3*, *ALPK3* and *FHOD3*) were also identified at genomic risk loci for DCM, a finding consistent with evidence that these disorders represent opposing extremes of a continuum of ventricular structure and systolic function[9,11]. We also identified *PITX2*, which has been previously shown to be strongly associated with atrial fibrillation (AF)[12]. To estimate the extent to which the DCM risk effects of PITX2, and the other identified risk loci, were related to AF, we conditioned the DCM GWAS summary statistics on AF using multitrait conditional and joint analysis (mtCOJO). Conditioning on AF partially attenuated the association signal at the *PITX2* locus, implying some genetic effects on DCM risk independent of AF. Genetic association estimates for all other loci were robust to conditional analysis on AF, suggesting that the genes identified primarily influence DCM risk (Extended Data Fig. 6).

Pathway analysis of prioritized genes identified enrichment of 72 cellular components and functions, including sarcomeric and cytoskeletal function, cellular adhesion and junction organization, aggrephagy, and Wnt and TGFβ signaling (Fig. 3b,c and Supplementary Table 9). Novel prioritized GWAS genes *MAPT*[13] and *MYL6* (ref. 14) contributed to the enrichment of pathways for contractile and cytoskeletal functions. The important role of cell-to-cell adhesion and cell-to-matrix interaction in DCM pathogenesis is underscored by the many effector genes acting at these interfaces. *STRN* encodes the desmosomal protein striatin, the canine ortholog of which has been implicated in dilated and arrhythmic cardiomyopathy[15]. *SSPN* encodes sarcospan, a key component of the dystrophin glycoprotein complex that has been linked to severe skeletal and cardiac muscle disorders. Other effector genes acting at the cell membrane identified include *MTSS1* (ref. 16), *PDLIM5* (refs. 17,18), *THBS1* and *TMEM182* (ref. 19).

Cell signaling components were prominently featured among the prioritized genes, including members of the TGFβ (*BAMBI*, *INHBB*, *PITX2* and *THBS1*) and Wnt (*CAMK2D*, *MAP3K7*, *NEDD4L*, *NFATC1*, *PRKCA* and *RNF207*) signaling pathways. *INHBB* encodes a secreted factor, and *THBS1* a transmembrane glycoprotein, both of which activate the TGFβ receptor, while *BAMBI* encodes a TGFβ-like pseudoreceptor that acts as a negative regulator of TGFβ signaling[20]. TGFβ signaling has been shown to be important in the development of fibrosis in cardiomyopathy models[21]. Several genes encoding heat-shock proteins (*HSPA4*, *HSPB7* and *HSPB8*) were also identified, expanding on the established role of *BAG3* and the unfolded protein response and endoplasmic reticular stress on DCM pathogenesis. Additionally, *FBXO32* encodes a muscle-specific ubiquitin ligase involved in protein degradation that has been proposed as a rare cause of DCM[22].

For genomic loci where a single high-confidence gene could not be identified, we manually curated the locus by integrating information from enriched biological pathways. The identified candidate genes were associated with cytoskeleton function (*ROCK2* (ref. 23) at locus 13), cell adhesion (*ITGA5* at locus 52), MAPK signaling (*EPHB1* at locus 23), and the unfolded protein response (*DNAJC18* at locus 31 and *CRYAB* at locus 50). Other notable genes included: the taurine transporter *SLC6A6* (locus 20), with existing evidence of taurine deficiency causing feline DCM[24]; the cardiac-expressed K+ channel *KCNIP2*, which has been implicated in Brugada syndrome and conduction abnormalities[25]; *RRAS2*, where gain of function variants are a cause of Noonan syndrome and accompanying hypertrophic cardiomyopathy[26,27]; and several genes implicated in myopathy, including *CHCHD10* (locus 80) and *DMPK* (locus 76).

## Rare variant burden association analysis of putative DCM effector genes

Within the identified DCM loci were seven Mendelian cardiomyopathy genes cataloged in ClinGen, a curated database of Mendelian-disease causing genes, with definitive evidence (DCM: *TTN*, *FLNC*, *LMNA*, *BAG3*; HCM: *MYBPC3*, *ALPK3*, *FHOD3*) and seven genes with moderate or limited evidence (DCM: *PRDM16*, *LDB3*; DCM or HCM: *OBSCN*, *VCL*, *NEXN*, *MYPN*; intrinsic cardiomyopathy: *ACTN2*). Emphasizing the role of gene dosage as a likely mechanism of action at GWAS genes[28] and the continuum of disease risk, four of the seven definitive evidence Mendelian DCM genes, established to act through mechanisms involving reduced gene product[29], were identified through GWAS: *TTN*, *FLNC*, *LMNA* and *BAG3*. We observed a tenfold enrichment of Mendelian cardiomyopathy genes within GWAS loci (odds ratio (OR) = 9.7, $P = 1.1 \times 10^{-6}$).

Next, we performed rare variant (MAF < 0.001) burden association analysis (RVAS), focusing on protein truncating variants (PTVs). This analysis was applied to (1) all DCM genes with definitive or moderate evidence for Mendelian DCM[30], to characterize the overall genetic architecture of DCM; and (2) genes prioritized at the identified GWAS loci through functional genomics analysis, to identify potential novel causes of Mendelian DCM and cardiomyopathy. In 453,455 participants with whole-exome sequencing from the UKB, a population-based cohort recruiting middle-aged and older individuals, the combined risk effects of rare variants in ClinGen definitive- or moderate-evidence DCM genes were orders of magnitude higher than those of GWAS sentinel variants mapping to the same genes (Fig. 4a and Supplementary Table 10).

To identify genes with a potential role in Mendelian DCM and cardiomyopathy, we investigated the effects of rare PTVs in the 62 prioritized genes with binary disease outcomes (cardiomyopathy and heart failure phenotypes) and quantitative CMR traits. Analysis was performed using whole-genome data in 78,142 individual participants of Genomics England (GeL), a rare disease and cancer cohort that recruited probands and their relatives from clinical centers, and with whole-exome sequencing in the UKB (including a subset of 36,104 with CMR). PTVs in three genes with limited or moderate evidence for Mendelian cardiomyopathy were nominally associated with DCM in GeL (*MYPN*: OR = 15.0, *P* = 0.03; *PRDM16*: OR = 40.3, *P* = 0.008) and with HCM in UKB (*NEXN*: OR = 24.1, *P* = 0.01) (Supplementary Tables 11 and 12). No carriers of *MYPN* or *PRDM16* PTVs where identified in UKB DCM cases, and only one case carried a *NEXN* PTV among HCM cases in GeL (OR = 1.3, *P* = 0.8) (Supplementary Tables 11 and 12). Rare PTVs in three prioritized genes, not established causes of cardiomyopathy, were found to be associated with binary diseases outcomes (*MAP3K7* and *NEDD4L* with DCM) in at least one cohort (Fig. 4b and Supplementary Tables 11 and 12) and with quantitative traits (*NEDD4L*, *MAP3K7* and *SSPN*) in UKB (Fig. 4b and Supplementary Table 13). PTVs in *MAP3K7* were associated with DCM in GeL (OR = 24.2, Benjamini–Hochberg adjusted *P* value ($P_{adj}$ = 0.02), and also with increased LV volumes (LV end-diastolic volume (LVEDV) = +54 ml, $P_{adj}$ = 0.01, LVESV = +38 ml,

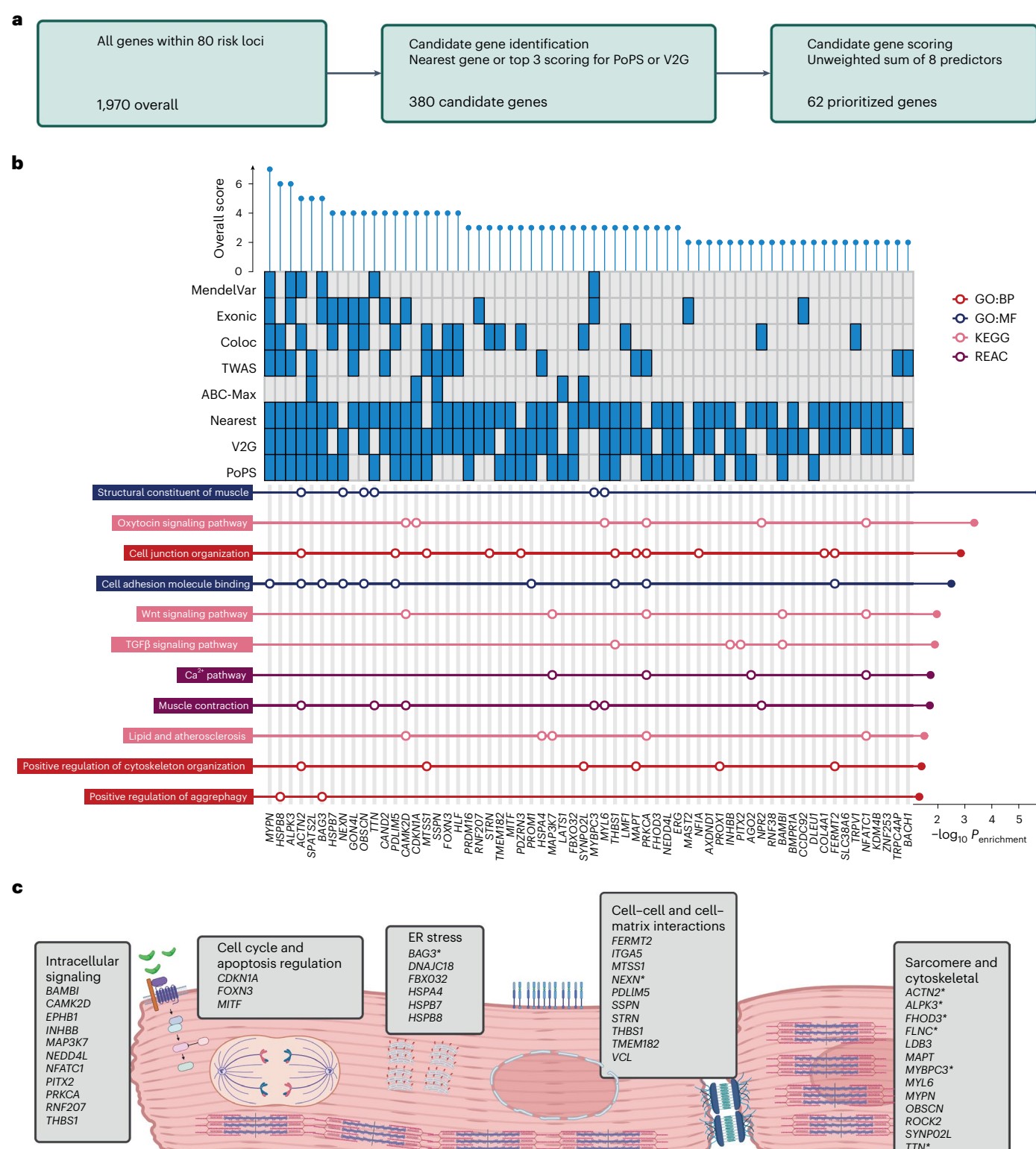

**Fig. 3 | Locus annotation and candidate gene scoring prioritize genes at risk loci and important biological pathways and processes in DCM pathogenesis.** **a**, Among all genes located within genomic risk loci (1,970 genes), candidate genes were selected based on proximity and being among the top three genes predicted using PoPS or V2G (380 candidate genes). Sixty-two genes were prioritized at 62 loci after scoring highest among the eight predictors. **b**, Pathway enrichment analysis of prioritized genes, highlighting pathways related to muscle structural constituents. Enrichment of effector genes within Gene Ontology pathways was performed using Fisher's one-sided test with Bonferroni adjustment of P values for the total number of pathways tested. **c**, Schematic overview of pathways and processes highlighted in DCM pathogenesis, manually curated from pathway enrichment analysis and published literature. Genes with existing evidence of being Mendelian causes of cardiomyopathy are highlighted in bold. Asterisk indicates moderate or definitive evidence of causing cardiomyopathy[30]. GO:BP, Gene Ontology: Biological Process; GO:MF, Gene Ontology: Molecular Function; KEGG, Kyoto Encyclopedia of Genes and Genomes; REAC (Reactome Pathway Database); ER, endoplasmic reticulum. **a** and **c** were created with BioRender.com.

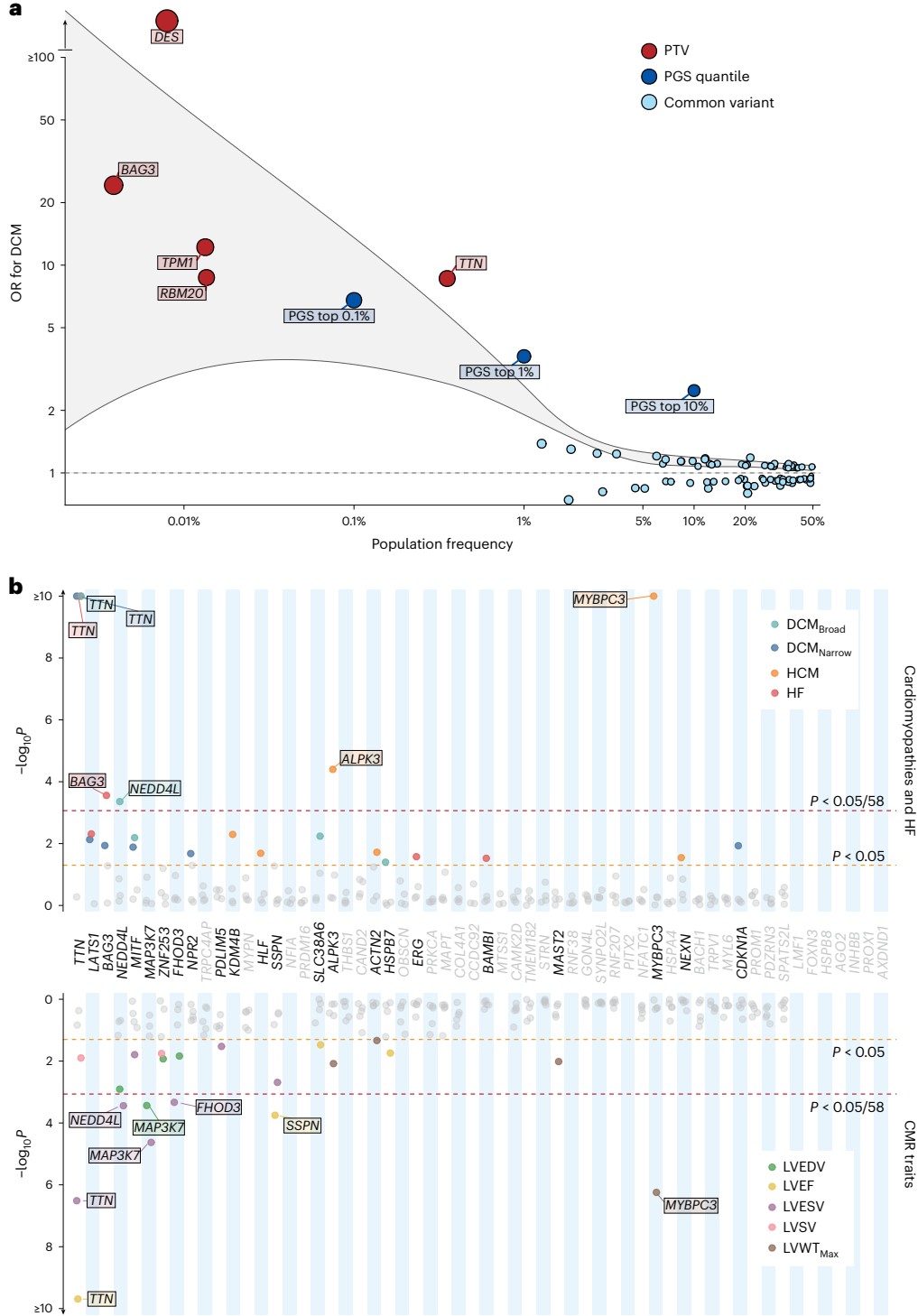

**Fig. 4 | Rare variant analysis highlights the genomic architecture of DCM and identifies novel disease- and trait-associated genes. a**, Genomic architecture of DCM incorporating effects arising from individual sentinel common variants (MAF > 0.01) in DCM loci (light blue), upper PGS quantiles of common variants (dark blue) and cumulative burden testing of rare PTVs (MAF < 0.001) in genes with moderate or definitive evidence of causing DCM[30] (red). Population frequency represents MAF for individual sentinel variants, the proportion of the population contained within the quantile for PGS, and the cumulative population frequency of rare variants in burden-tested genes. Outcome for burden testing was DCM, with presentation of all genes reaching nominal significance (P < 0.05) following logistic ridge regression with Firth correction implemented using REGENIE. The gray highlighted region indicates smoothened regression lines

of the upper and lower bounds for each effect estimate. **b**, Burden analysis of rare PTVs (MAF < 0.001) in 58 prioritized protein-coding genes in UKB (453,455 participants with whole-exome sequencing, and 36,104 with CMR), highlighting established Mendelian cardiomyopathy genes (*TTN*, *BAG3*, *FHOD3*, *ALPK3* and *MYBPC3*) and three novel genes (*NEDD4L*, *MAP3K7* and *SSPN*). Red line indicates statistical significance (P < 8.6 × 10⁻⁴; 0.05 of 58 genes), and orange line indicates nominal significance (P < 0.05). Genes are ordered by mean P value across all tested traits, from lowest to highest, with genes reaching nominal significance (P < 0.05) for at least one trait highlighted in bold. Burden testing was performed using logistic ridge regression with Firth correction implemented using REGENIE. Detailed results are available in Supplementary Tables 11–13. HF, heart failure; LVSV, LV stroke volume; LVWTM_{ax}, maximum LV wall thickness.

$P_{adj} = 4.4 \times 10^{-4}$) in UKB. The importance of *MAP3K7* in DCM pathogenesis was futher underscored by the prioritization of additional pathway genes, including *RNF207* (ref. 31), a regulator of *MAP3K7* activation, which has been identified as a possible cause of canine DCM[32]. PTVs in membrane receptor regulator *NEDD4L* were associated with DCM (OR = 10.4, $P_{adj}$ = 0.01) *P* and with quantitative traits in UKB (PTV: LVEDV = +29.7, $P_{adj}$ = 0.02; LVESV = +19.8, $P_{adj}$ = 0.005), with replication in GeL (heart failure OR = 13.0, *P* = 0.01). PTVs in *SSPN* were associated with significant changes in quantitative LV traits (LVEF −5.9%, $P_{adj}$ = 0.004 and LVESV + 13.0 ml, $P_{adj}$ = 0.02). Within a local DCM cohort, three of 337 cases (0.9%) carried PTVs in *SSPN*, compared with 80 of 352,564 (0.02%) among UKB controls ($P = 1 \times 10^{-5}$). *SSPN* is a critical protein located within the dystrophin glycoprotein complex of muscle cells, including cardiomyocytes. Its activity protects against impairment of cardiac contractility resulting from dystrophin deficiency in Duchenne muscular dystrophy, whereas loss of function destabilizes muscle adhesion and force generation[33,34]. An exploratory analysis of ultrarare variants (MAF < $1 \times 10^{-5}$) that did not meeting the minor allele threshold in UKB for the main RVAS, identified additional associations with DCM, specifically with *SLC38A6* and *SSPN* (Supplementary Table 14).

## Identifying key cell types and cellular processes using single-cell transcriptomics

To identify the organs, tissues and cell types mediating genetic risk of DCM, we performed bulk tissue-level heritability enrichment analysis. Cardiac and other muscle-related tissues (including vascular and gastrointestinal smooth muscle) showed the highest levels of enrichment (Fig. 5a and Supplementary Table 15). Cell type heritability was assessed using the sc-linker framework[35], integrating single-nucleus RNA sequencing (snRNA-seq)[36] of LV tissue from 52 DCM patients with end-stage heart failure undergoing cardiac transplantation and 18 controls, and genome-wide enhancer–promoter contact in the LV, with GWAS heritability. We identified biologically relevant cell types and disease-specific relationships by identifying enrichments in basal gene expression profiles within cardiomyocytes and DCM-specific differentially expressed genes (DEGs) in cardiomyocytes, fibroblasts and mural cells (Fig. 5b and Supplementary Tables 16 and 17). When gene expression in control hearts was evaluated, most prioritized genes had the highest levels of expression in cardiomyocytes (Fig. 5c). Several of the prioritized DCM genes, including *SSPN*, *MAP3K7* and *NEDD4L*, were differentially expressed in cardiomyocytes in DCM (Fig. 5d). Supporting the important role of noncardiomyocytes in DCM pathogenesis, fibroblasts and mural cells (primarily pericytes) consistently had higher proportions of DEGs in enriched biological pathways (Extended Data Fig. 7), with most prioritized genes being DEGs in noncardiomyocytes.

To explore cardiomyocyte and cardiomyocyte cell-nonautonomous mechanisms, as well as the role of prioritized genes encoding ligands or receptors, we investigated intercellular signaling pathways using CellChat[37]. This method combines cellular transcriptomics, a priori knowledge of ligand–receptor–cofactor interactions and the law of mass action to quantify communication networks. In DCM, we observed an overall increase in global signaling, with notable reductions in cardiomyocyte–cardiomyocyte interaction strength (Extended Data Fig. 7). Additionally, there was an increase in interactions of prioritized genes enriched in the TGFβ signaling pathway, along with specific changes in pathways containing specific prioritized genes. For example, interactions of *COL4A1* and *EPHB1* increased, while those of *THBS1* decreased (Extended Data Fig. 7). Modest increases in overall collagen signaling were also found in DCM. Specifically, *COL4A1* expression was increased in fibroblasts (Fig. 5d), with enhanced signaling to cardiomyocytes, fibroblasts and mural cells via integrins (Fig. 5e). *EPHB1* (encoding Ephrin type-B receptor 1) expression was highest in cardiomyocytes, while its cognate ligand, EFNB2 (encoding Ephrin-B2), was expressed in endothelial cells. In DCM, the levels of the ligand increased, while there was a corresponding decrease in receptor production (Extended Data Fig. 7). Similar findings were reported in a single-nucleus RNA-sequencing study of pressure-overloaded human hearts[38]. *BMPR1A* was predominantly expressed in cardiomyocytes (Extended Data Fig. 7), with increased expression in mural cells and fibroblasts. This was associated with increased *BMP6*–*BMPR1A* signaling from endocardial cells to cardiomyocytes and fibroblasts (Fig. 5f and Extended Data Fig. 7), as previously reported[36].

## Polygenic burden predicts risk and modifies penetrance in carriers of monogenic variants

Given the important contribution of common genetic variation to DCM heritability, we generated a polygenic score (PGS_DCM) using 541,841 SNP predictors and evaluated it in 347,585 unrelated participants of White British ancestry from the UKB (Fig. 6a). The PGS was significantly associated with DCM (OR per PGS s.d. 1.76, 95% CI 1.64 to 1.90, $P < 2 \times 10^{-16}$; area under the receiver operating characteristic curve (AUROC) = 0.71) in the general population. The top centile had a fourfold increased risk compared with the median (OR = 3.83, 95% CI 2.52 to 5.79, $P = 2.1 \times 10^{-10}$), and a sevenfold increased risk compared with the bottom centile (OR = 7.04, 95% CI 2.42 to 20.52, $P = 3.5 \times 10^{-4}$) (Fig. 6b,c). In 25,443 individuals from the UKB with CMR imaging, PGS_DCM was associated with cardiac traits concordant with DCM (Supplementary Table 18). These included reduced contractility (LVEF: per PGS s.d. −0.7%, $P_{adj} = 8.1 \times 10^{-78}$; top versus bottom centile 57.6 versus 60.8, $P_{adj} = 1.7 \times 10^{-6}$) and increased volumes (LVEDV: +2.1 ml, $P_{adj} = 2.5 \times 10^{-45}$; top versus bottom centile: 158.1 versus 143.4, $P = 3.1 \times 10^{-6}$; LVESV: +1.9, $P = 1.6 \times 10^{-93}$; top versus bottom centile: 67.7 versus 56.6, $P = 1.4 \times 10^{-9}$). Given the variability in penetrance and expressivity of DCM in carriers of rare pathogenic variants[39], we next evaluated whether common variants affected penetrance of rare variants, as has previously been demonstrated in HCM[11]. In 1,546 carriers of pathogenic variants in DCM-causing genes in UKB (prevalence 0.5%), PGS_DCM stratified DCM prevalence (top quintile: 7.3%,

**Fig. 5 | Integration of genomics and transcriptomics identifies genes and biological mechanisms in DCM. a,b,** Partitioned heritability at tissue level (**a**) and at cell type level (**b**) from snRNA-seq data of 52 DCM cases and 18 controls. Enrichment *P* values were adjusted using the Benjamini–Hochberg method. Dashed line indicates FDR-adjusted *P* value of 0.05. For cell-type-specific heritability enrichment, cardiomyocyte marker and disease-specific expression in cardiomyocytes and mural cell types remained significant when the tau coefficient was used (Supplementary Table 16). **c,** Cell type expression of prioritized genes in single-nucleus transcriptomics from LV tissue in 18 control donors. Mean expression is scaled from minimum to maximum, and the proportion of expressing nuclei within a cell type is indicated by dot size. Cardiomyocyte expression is indicated in the gray shaded box. **d,** Differential expression of candidate genes across the range of major cell types. Red and blue indicate increased and reduced gene expression in DCM compared with controls, respectively. Yellow dot indicates significant DEGs within a cell type at FDR < 0.05. Genes are ordered by highest absolute log fold-change difference across cell types. Cell types are ordered by abundance from greatest (outer) to least (inner). **e,** Increased *COL4A1* signaling from fibroblasts to cardiomyocytes, fibroblasts and mural cells via integrins from DCM single-nucleus transcriptomics. Communication probability indicates the scaled strength of interaction from maximum to minimum signaling interactions between cell types. Dot color reflects communication probabilities, and dot size represents *P* values computed by one-sided permutation test. **f,** Upregulation of *BMP6* (ligand) in endocardial cells, resulting in increased signaling through *BMPR1A* in cardiomyocytes, fibroblasts and mural cells. Communication probability indicates the scaled strength of interaction from maximum to minimum signaling interactions between cell types. Dot color reflects communication probabilities, and dot size represents *P* values computed by one-sided permutation test. NC, neuronal cell; AD, adipocyte; FC, fold change; CNS, central nervous system; Max., maximum; Min., minimum.

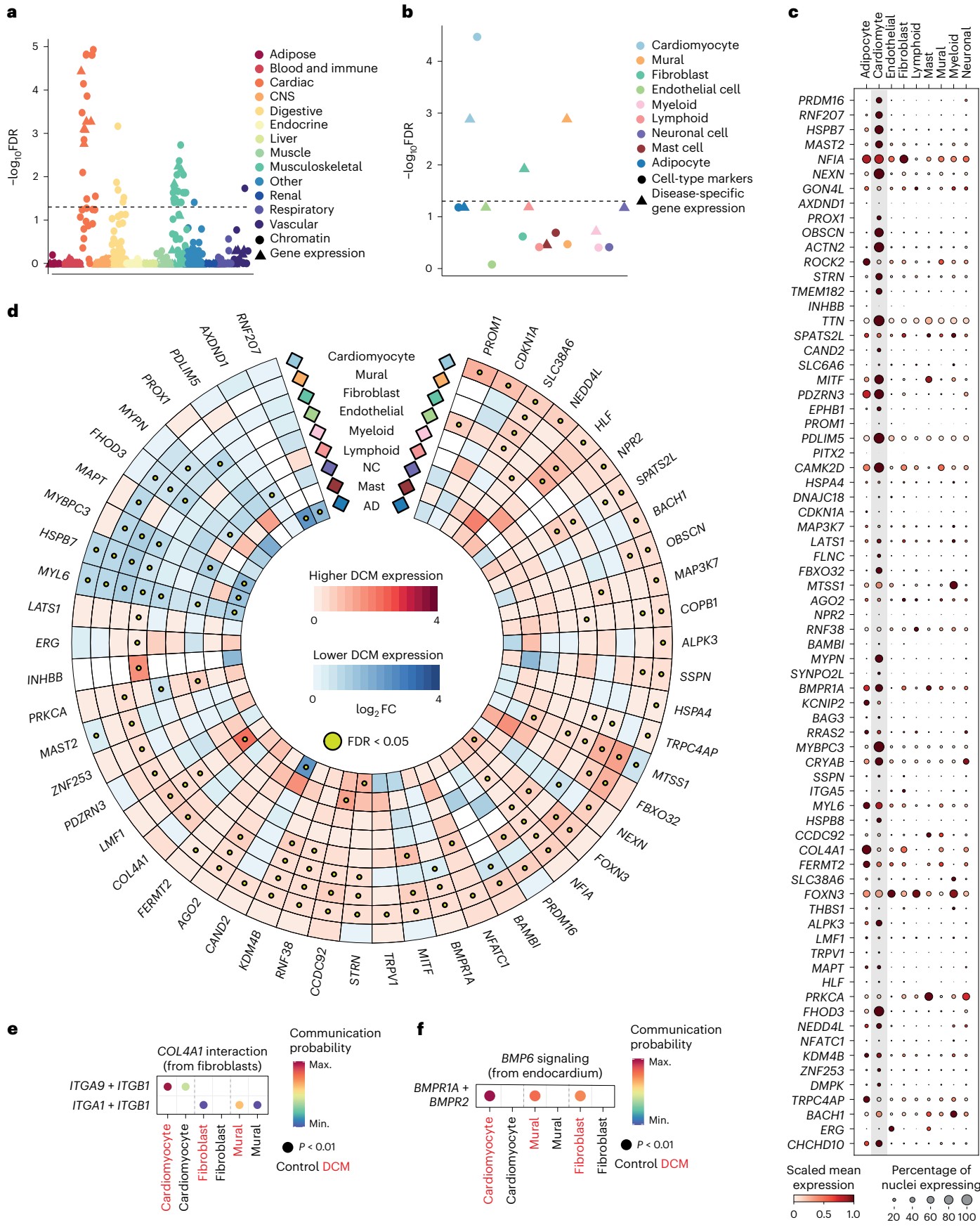

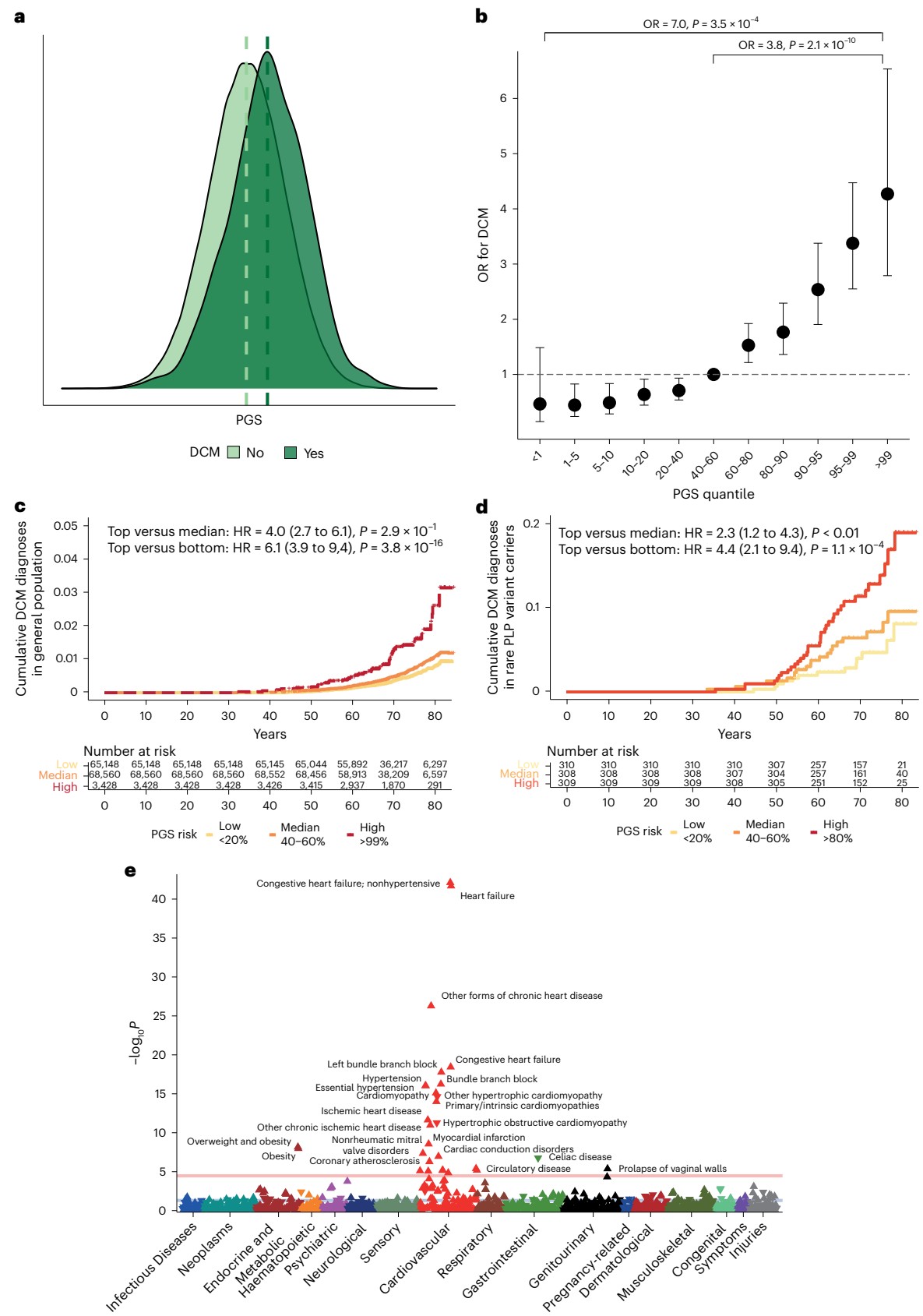

bottom quintile: 1.7%, $P$ 0.005), including in 1,166 carriers of rare *TTN* PTVs (Fig. 6d). DCM risk was higher in carriers of pathogenic variants in DCM-causing genes compared with gene-negative individuals in the top centile of PGS risk (OR = 6.4, 95% CI 4.0 to 10.3, $P = 6 \times 10^{-14}$).

Finally, we conducted a phenome-wide association study (pheWAS) of $PGS_{DCM}$ to explore genetic relationships between common variant risk and other traits and diseases. We identified significant associations with heart failure and several related cardiovascular phenotypes

**Fig. 6 | DCM PGS is associated with DCM disease status in the UKB, including in carriers of pathogenic or likely pathogenic variants in DCM-causing genes.** **a**, PGS distribution among 347,585 UKB participants with and without DCM, showing higher PGS in those with DCM. **b**, ORs and 95% confidence intervals for DCM in quantile bins among 347,585 UKB participants, comparing individuals in the top centile (n = 3,428) with those in the median 40–60% centiles (n = 68,560) and lowest centile (n = 3,428). *P* values are two-sided and were calculated from a logistic regression model and not adjusted for multiple testing. **c**, Cumulative hazards for lifetime diagnosis of DCM in the UKB stratified by high (top 1%, red), median (middle 20%, orange) and low (bottom 20%, yellow) PGS. *P* values are two-sided and were calculated from a Cox proportional hazards regression model and not adjusted for multiple testing. **d**, Cumulative hazards for lifetime diagnosis of DCM in carriers of pathogenic or likely pathogenic (PLP) rare

variants in DCM-causing genes in UKB, stratified by high (top 20%, red), median (middle 20%, orange) and low (bottom 20%, yellow) PGS. *P* values are two-sided and were calculated from a Cox proportional hazards regression model and not adjusted for multiple testing. **e**, Manhattan plot of DCM PGS pheWAS in UKB, showing associations with cardiovascular phenotypes and obesity. ICD-9 and ICD-10 diagnostic codes are mapped to PheCode Map v.1.2. Mapped phenotypes exceeding the phenome-wide significance threshold ($P = 2.7 \times 10^{-5}$, red line, adjusted for the total number of tested phenotypes) are labeled. The blue line indicates the nominal significance level ($P < 0.05$). The direction of the triangle indicates the direction of effect of the PGS association. *P* values are two-sided and were calculated from the linear regression model and not adjusted for multiple testing. PheWAS analyses adjusted for DCM or heart failure and hypertension status are shown in Extended Data Fig. 8. HR, hazard ratio.

(electrophysiologic and valvular), as well as established risk factors for impaired cardiac function (hypertension and obesity) (Fig. 6e). We also found significant associations with cardiac ischemic phenotypes and inverse associations with HCM, as previously described[9]. Genetic association estimates for all DCM loci were robust to conditional analysis on CAD and systolic blood pressure (SBP) using mtCOJO, suggesting that the identified genes primarily affect DCM risk (Extended Data Fig. 6). The pheWAS associations were robust to adjustment for measured hypertension, while adjustment for DCM and heart failure diagnoses resulted in loss of associations with ischemic phenotypes and obesity (Extended Data Fig. 8).

## Discussion

In conclusion, through GWAS meta-analysis and multitrait analysis with LV traits, we identified 59 genomic loci for novel DCM, 31 of which had not been previously reported. These loci, along with an additional 21 loci significant at an FDR of 1% (80 loci in total), were investigated using a systematic approach for locus annotation and gene prioritization. We prioritized 62 effector genes for DCM, which were associated with key biological pathways in disease pathogenesis. Using single-nucleus transcriptomics from explanted end-stage DCM hearts, we demonstrated the importance of these pathways and highlighted the key role of noncardiomyocyte cell types and noncell-autonomous effects, including Ephrin-B and BMP6 signaling. Rare variant association analysis of the prioritized genes also identified previously unrecognized potential causes of Mendelian DCM, including *MAP3K7*, *NEDD4L* and *SSPN*. Finally, we demonstrate that a DCM polygenic score directly affects DCM risk and modifies disease penetrance in carriers of rare pathogenic variants. These findings provide mechanistic insights into the genetic architecture and molecular etiology of DCM and may inform therapeutic strategies for both DCM patients and at-risk individuals.

## Online content

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

Sean L. Zheng [1,2,3,72], Albert Henry [4,5,72], Douglas Cannie[4,6], Michael Lee [1], David Miller[7], Kathryn A. McGurk [1,2,13], Isabelle Bond[4], Xiao Xu[1,2], Hanane Issa[5], Catherine Francis[1,3], Antonio De Marvao[1,2,3], Pantazis I. Theotokis [1,2,3], Rachel J. Buchan [1,2,3], Doug Speed [8], Erik Abner [9], Lance Adams[10], Krishna G. Aragam [11,12,13], Johan Ärnlöv[14,15], Anna Axelsson Raja[16], Joshua D. Backman [17], John Baksi[3], Paul J. R. Barton [1,2,3], Kiran J. Biddinger[11,13], Eric Boersma[18], Jeffrey Brandimarto[19], Søren Brunak [20], Henning Bundgaard[16], David J. Carey[21], Philippe Charron [22,23], James P. Cook[24], Stuart A. Cook[1,2,3], Spiros Denaxas[5,25,26,27], Jean-François Deleuze [28,29,30], Alexander S. Doney[31], Perry Elliott [4,6], Christian Erikstrup [32,33], Tõnu Esko[9,13], Eric H. Farber-Eger [34], Chris Finan [4], Sophie Garnier[22], Jonas Ghouse[16], Vilmantas Giedraitis [35], Daniel F. Guðbjartsson [36,37], Christopher M. Haggerty[21], Brian P. Halliday[1,3], Anna Helgadottir [36], Harry Hemingway [5,25], Hans L. Hillege[38], Isabella Kardys[18], Lars Lind[39], Cecilia M. Lindgren[13,40,41], Brandon D. Lowery [34], Charlotte Manisty[4,6], Kenneth B. Margulies[20], James C. Moon[4,6], Ify R. Mordi[31], Michael P. Morley[20], Andrew D. Morris [42], Andrew P. Morris[43], Lori Morton[44], Mahdad Noursadeghi[45], Sisse R. Ostrowski [46,47], Anjali T. Owens[19], Colin N. A. Palmer[48], Antonis Pantazis[3], Ole B. V. Pedersen [47,49], Sanjay K. Prasad [1,3], Akshay Shekhar[44], Diane T. Smelser[21], Sundararajan Srinivasan [48], Kari Stefansson[36,50], Garðar Sveinbjörnsson [36], Petros Syrris [4], Mari-Liis Tammesoo[9], Upasana Tayal[1,3], Maris Teder-Laving[9], Guðmundur Thorgeirsson [36,50], Unnur Thorsteinsdottir[36,50], Vinicius Tragante [36], David-Alexandre Trégouët [29,51], Thomas A. Treibel[4,6], Henrik Ullum[52], Ana M. Valdes[53], Jessica van Setten [54], Marion van Vugt[54], Abirami Veluchamy[48], W. M. Monique Verschuren [55,56], Eric Villard [22], Yifan Yang[19], COVIDsortium, DBDS Genomic Consortium, Estonian Biobank Research Team, HERMES Consortium*, Folkert W. Asselbergs[4,27,57], Thomas P. Cappola [19], Marie-Pierre Dube [58,59], Michael E. Dunn[44], Patrick T. Ellinor [13,60], Aroon D. Hingorani [4], Chim C. Lang[31,61], Nilesh J. Samani[62], Svati H. Shah[63,64,65], J. Gustav Smith [66,67,68], Ramachandran S. Vasan [69,70], Declan P. O'Regan[2], Hilma Holm [36], Michela Noseda [1], Quinn Wells[71], James S. Ware [1,2,3,13,73] ✉ & R. Thomas Lumbers [5,25,26,73] ✉

[1]National Heart and Lung Institute, Imperial College London, London, UK. [2]MRC Laboratory of Medical Sciences, London, UK. [3]Royal Brompton & Harefield Hospitals, Guy's and St. Thomas' NHS Foundation Trust, London, UK. [4]Institute of Cardiovascular Science, University College London, London, UK. [5]Institute of Health Informatics, University College London, London, UK. [6]Barts Heart Centre, St Bartholomew's Hospital, London, UK. [7]Division of Biosciences, University College London, London, UK. [8]Quantitative Genetics and Genomics, Aarhus University, Aarhus, Denmark. [9]Estonian Genome Center, Institute of Genomics, University of Tartu, Tartu, Estonia. [10]Geisinger Health System, Danville, PA, USA. [11]Cardiovascular Research Center, Massachusetts General Hospital, Boston, MA, USA. [12]Center for Genomic Medicine, Massachusetts General Hospital, Boston, MA, USA. [13]Program in Medical and Population Genetics, The Broad Institute of MIT and Harvard, Cambridge, MA, USA. [14]Department of Neurobiology, Care Sciences and Society/Section of Family Medicine and Primary Care, Karolinska Institutet, Stockholm, Sweden. [15]School of Health and Social Sciences, Dalarna University, Falun, Sweden. [16]Department of Cardiology, The Heart Centre, Copenhagen University Hospital, Rigshospitalet, Copenhagen, Denmark. [17]Analytical Genetics, Regeneron Genetics Center, Tarrytown, NY, USA. [18]Erasmus MC, Cardiovascular Institute, Thorax Center, Department of Cardiology, Utrecht, the Netherlands. [19]Penn Cardiovascular Institute, Perelman School of Medicine, University of Pennsylvania, Philadelphia, PA, USA. [20]Novo Nordisk Foundation Center for Protein Research, Faculty of Health and Medical Sciences, University of Copenhagen, Copenhagen, Denmark. [21]Department of Molecular and Functional Genomics, Geisinger, Danville, PA, USA. [22]Sorbonne Research Unit on Cardiovascular Disorders, Metabolism and Nutrition, Team Genomics & Pathophysiology of Cardiovascular Diseases, ICAN Institute for Cardiometabolism and Nutrition, Paris, France. [23]APHP, Department of Genetics, Pitié-Salpêtrière Hospital, Paris, France. [24]Department of Biostatistics, University of Liverpool, Liverpool, UK. [25]Health Data Research UK, University College London, London, UK. [26]British Heart Foundation Data Science Centre, London, UK. [27]The National Institute for Health Research University College London Hospitals Biomedical Research Centre, University College London, London, UK. [28]Centre National de Recherche en Génomique Humaine (CNRGH), Institut de Biologie François Jacob, CEA, Université Paris-Saclay, Evry, France. [29]Laboratory of Excellence GENMED (Medical Genomics), Paris, France. [30]Centre d'Etude du Polymorphisme Humain, Fondation Jean Dausset, Paris, France. [31]Division of Molecular & Clinical Medicine, University of Dundee, Ninewells Hospital and Medical School, Dundee, UK. [32]Department of Clinical Immunology, Aarhus University Hospital, Aarhus, Denmark. [33]Deparment of Clinical Medicine, Aarhus University Hospital, Aarhus, Denmark. [34]Vanderbilt Institute for Clinical and Translational Research, Vanderbilt University Medical Center, Nashville, TN, USA. [35]Department of Public Health and Caring Sciences, Geriatrics, Uppsala, Sweden. [36]deCODE genetics/Amgen Inc., Reykjavik, Iceland. [37]School of Engineering and Natural Sciences, University of Iceland, Reykjavik, Iceland. [38]Department of Cardiology, University Medical Center Groningen, University of Groningen, Groningen, the Netherlands. [39]Department of Medical Sciences, Uppsala University, Uppsala, Sweden. [40]Big Data Institute at the Li Ka Shing Centre for Health Information and Discovery, University of Oxford, Oxford, UK. [41]Wellcome Trust Centre for Human Genetics, University of Oxford, Oxford, UK. [42]Usher Institute of Population Health Sciences and Informatics, University of Edinburgh, Edinburgh, UK. [43]Centre for Genetics and Genomics Versus Arthritis, Centre for Musculoskeletal Research, University of Manchester, Manchester, UK. [44]Cardiovascular Research, Regeneron Pharmaceuticals, Tarrytown, NY, USA. [45]Research Department of Infection, Division of Infection and Immunity, University College London, London, UK. [46]Department of Clinical Immunology, Copenhagen University Hospital, Rigshospitalet, Copenhagen, Denmark. [47]Department of Clinical Medicine, Faculty of Health and Medical Sciences, University of Copenhagen University Hospital, Copenhagen, Denmark. [48]Division of Population Health and Genomics, University of Dundee, Ninewells Hospital and Medical School, Dundee, UK. [49]Department of Clinical Immunology, Zealand University Hospital, Køge, Denmark. [50]Department of Medicine, University of Iceland, Reykjavik, Iceland. [51]Univ. Bordeaux, INSERM, BPH, Bordeaux, France. [52]Statens Serum Institut, Copenhagen, Denmark. [53]Injury, Recovery and Inflammation Sciences, School of Medicine, University of Nottingham, Nottingham, UK. [54]Department of Cardiology, University Medical Center Utrecht, Utrecht, the Netherlands. [55]Department Life Course, Lifestyle and Health, Centre for Prevention, Lifestyle and Health, National Institute for Public Health and the Environment, Bilthoven, the Netherlands. [56]Julius Center for Health Sciences and Primary Care, University Medical Center Utrecht, Utrecht, the Netherlands. [57]Department of Cardiology, Amsterdam University Medical Centers, Amsterdam, the Netherlands. [58]Montreal Heart Institute, Montreal Heart Institute, Montreal, Quebec, Canada. [59]Faculty of Medicine, Université de Montréal, Montreal, Quebec, Canada. [60]Cardiac Arrhythmia Service and Cardiovascular Research Center, Massachusetts General Hospital, Boston, MA, USA. [61]Tuanku Muhriz Chair, Universiti Kebangsaan Malaysia, Kuala Lumpur, Malaysia. [62]Department of Cardiovascular Sciences, University of Leicester and NIHR Leicester Biomedical Research Centre, Glenfield Hospital, Leicester, UK. [63]Department of Medicine, Division of Cardiology, Duke University Medical Center, Durham, NC, USA. [64]Duke Clinical Research Institute, Durham, NC, USA. [65]Duke Molecular Physiology Institute, Durham, NC, USA. [66]Department of Cardiology, Clinical Sciences, Lund University and Skåne University Hospital, Lund, Sweden. [67]Department of Molecular and Clinical Medicine, Institute of Medicine, Gothenburg University and Sahlgrenska University Hospital, Gothenburg, Sweden. [68]Wallenberg Center for Molecular Medicine and Lund University Diabetes Center, Lund University, Lund, Sweden. [69]National Heart, Lung, and Blood Institute's and Boston University's Framingham Heart Study, Framingham, MA, USA. [70]Sections of Cardiology, Preventive Medicine and Epidemiology, Department of Medicine, Boston University Schools of Medicine and Public Health, Boston, MA, USA. [71]Division of Cardiovascular Medicine, Vanderbilt University, Nashville, TN, USA. [72]These authors contributed equally: Sean L. Zheng, Albert Henry. [73]These authors jointly supervised this work: James S. Ware, R. Thomas Lumbers. ✉e-mail: j.ware@imperial.ac.uk; t.lumbers@ucl.ac.uk

**COVIDsortium**

Aroon D. Hingorani[4], Charlotte Manisty[4,6], James C. Moon[4,6], Mahdad Noursadeghi[44] & Thomas A. Treibel[4,6]

**DBDS Genomic Consortium**

Søren Brunak[20], Christian Erikstrup[32,33], Daniel F. Guðbjartsson[36,37], Ole B. V. Pedersen[46,48], Kari Stefansson[36,49], Unnur Thorsteinsdottir[36,49] & Henrik Ullum[51]

**Estonian Biobank Research Team**

Erik Abner[9] & Tõnu Esko[9,13]

## HERMES Consortium

Erik Abner[9], Lance Adams[10], Krishna G. Aragam[11,12,13], Johan Ärnlöv[14,15], Folkert W. Asselbergs[4,27,56], Joshua D. Backman[17], John Baksi[3], Paul J. R. Barton[1,2,3], Kiran J. Biddinger[11,13], Eric Boersma[18], Isabelle Bond[4], Jeffrey Brandimarto[19], Rachel J. Buchan[1,2,3], Douglas Cannie[4,6], Thomas P. Cappola[19], David J. Carey[21], Philippe Charron[22,23], James P. Cook[24], Stuart A. Cook[1,2,3], Antonio De Marvao[1,2,3], Spiros Denaxas[5,25,26,27], Alexander S. Doney[31], Marie-Pierre Dube[57,58], Michael E. Dunn[43], Patrick T. Ellinor[13,59], Perry Elliott[4,6], Tõnu Esko[9,13], Eric H. Farber-Eger[34], Chris Finan[4], Catherine Francis[1,3], Sophie Garnier[22,23], Jonas Ghouse[16], Vilmantas Giedraitis[35], Daniel F. Guðbjartsson[36,37], Christopher M. Haggerty[21], Brian P. Halliday[1,3], Anna Helgadottir[36], Harry Hemingway[5,25], Albert Henry[4,5,72], Hans L. Hillege[38], Aroon D. Hingorani[4], Hilma Holm[36], Hanane Issa[5], Isabella Kardys[18], Chim C. Lang[31,61], Michael Lee[1], Lars Lind[39], Cecilia M. Lindgren[13,40,41], Brandon D. Lowery[34], R. Thomas Lumbers[5,25,26,73], Kenneth B. Margulies[20], Kathryn A. McGurk[1,2,13], David Miller[7], Ify R. Mordi[31], Michael P. Morley[20], Andrew D. Morris[42], Andrew P. Morris[24,41], Lori Morton[43], Michela Noseda[1], Declan P. O'Regan[2], Anjali T. Owens[19], Colin N. A. Palmer[47], Antonis Pantazis[3], Sanjay K. Prasad[1,3], Nilesh J. Samani[60], Svati H. Shah[61,62,63], Akshay Shekhar[46], Diane T. Smelser[21], J. Gustav Smith[64,65,66], Doug Speed[8], Sundarajan Srinivasan[47], Kari Stefansson[36,49], Garðar Sveinbjörnsson[36], Petros Syrris[4], Mari-Liis Tammesoo[9], Upasana Tayal[1,3], Maris Teder-Laving[9], Pantazis I. Theotokis[1,2,3], Guðmundur Thorgeirsson[36,49], Unnur Thorsteinsdottir[36,49], Vinicius Tragante[36], Ramachandran S. Vasan[67,68], Jessica van Setten[53], Marion van Vugt[53], Abirami Veluchamy[47], W. M. Monique Verschuuren[54,55], Eric Villard[22,23], James S. Ware[1,2,3,13,73], Quinn Wells[69], Xiao Xu[1,2], Yifan Yang[19] & Sean L. Zheng[1,2,3,72]

A full list of members and their affiliations appears in the Supplementary Information

## Methods

### Ethics statement

This research complied with all relevant ethical regulations. All patients gave written informed consent, and all studies were approved by the relevant regional research ethics committees and adhered to the principles set out in the Declaration of Helsinki. Details of ethics approvals for individual studies are provided in the Supplementary Information.

### Phenotype and study populations

DCM was defined in each participating study using a harmonized, rule-based, multimodal phenotyping algorithm as a guide. DCM was defined as LV systolic dysfunction with or without LV dilatation in the absence of secondary causes of heart failure (CAD, valvular heart disease or congenital heart disease); see Supplementary Information 1 for full definitions. Individuals with CAD, valvular heart disease or congenital heart disease were excluded from the control group. Imaging evidence or physician adjudication was preferred, but, where this was unavailable, classifiers were defined as the presence of at least one relevant diagnosis or procedural code from the patient's medical records.

### Discovery GWAS and multitrait analysis of GWAS

The current GWAS meta-analysis included 14,256 cases and 1,199,156 controls of European ancestry from 16 studies in the HERMES Consortium (cohorts described in Supplementary Information 2 and Supplementary Table 1). Genotyping for 15 of 16 studies was performed locally in each participating study using high-density genotyping arrays imputed against reference whole-genome sequencing panels from the Haplotype Reference Consortium (14 studies), 1000 Genomes Project (ref. 40) or population-specific reference panels (Estonian Biobank and deCODE) (Supplementary Table 2). Genotyping for the GeL cohort was done using whole-genome sequencing. Genetic association tests were performed per study per phenotype, using a logistic regression model assuming additive genetic effects with adjustments for age, sex, genetic principal components (PCs) and study-specific covariates. Full details of study-level GWAS methods are available in Supplementary Information 3 and Supplementary Table 2. Descriptions of studies and participant characteristics are provided in Supplementary Table 1. Sensitivity analysis GWAS and meta-analysis of strictly defined DCM (Supplementary Information 1) were performed using the same workflow. To assess the effects of ascertainment of DCM using the different criteria, GWAS meta-analysis was performed for the studies that used narrow ($DCM_{Narrow}$ GWAS) or broad ($DCM_{Broad}$ GWAS) criteria (Supplementary Table 1), and genetic correlations were assessed using bivariate LD score regression with LDSC v.1.0.1 (ref. 41).

GWAS meta-analysis was performed centrally using METAL v.2020-05-05 (ref. 42) with an inverse-variance weighted fixed-effect model. To boost discovery power, we further conducted a multitrait analysis of GWAS (MTAG), a method for jointly analyzing summary statistics from multiple overlapping GWAS of genetically correlated traits. GWAS in the UK Biobank of ten CMR-derived LV traits (LVEF, LVESV, LVEDV, stroke volume, global circumferential, longitudinal and radial strains, mass, concentricity, and maximum wall thickness) from 36,083 unrelated participants of White British ancestry and without heart failure, cardiomyopathy, previous myocardial infarction or structural heart disease[8] were tested for genetic correlation with primary GWAS using LDSC v.1.0.1 (refs. 43,44). MTAG of the primary GWAS was then performed with CMR traits with high genetic correlation ($|r_g| > 0.7$) using mtag v.1.0.8 (ref. 10). The maximum FDR was estimated by mtag to be 2.7%.

### SNP-based heritability estimation

The proportion of variance in heart failure risk explained by common SNPs — that is, SNP-based heritability $(h^2_{SNP})$ — was estimated from GWAS meta-analysis summary statistics using LDAK SumHer software v.5.2 with the BLD-LDAK heritability model[7]. The $h^2_{SNP}$ estimates were calculated on a liability scale, which assumes that a binary phenotype has an underlying continuous liability, and that above a certain liability threshold, an individual becomes affected[45]. To model the expected heritability tagged by each SNP, we used precomputed tagging files derived from 2,000 White British individuals, and we used a correction for sample prevalence by calculating the effective sample size assuming equal numbers of cases and controls[46]. The conversion to liability scale was calculated using a population prevalence of 0.004 for $DCM_{Narrow}$ (based on an estimated prevalence of 1 in 250 individuals[2,3]) and 0.008 for DCM (assuming twice the prevalence of $DCM_{Narrow}$).

### Locus identification

To identify genetic susceptibility loci for DCM, we first identified conditionally independent genetic variants using a chromosome-wide stepwise conditional-joint analysis implemented in the Genome-wide Complex Trait Analysis software (v.1.92.4)[47] at a genome-wide significance threshold of $P < 5 \times 10^{-8}$ in all GWAS and additionally at FDR < 1% (estimated using qvalue) for DCM GWAS. To define a genomic locus, conditionally independent genetic variants across both DCM GWAS and DCM MTAG that were located within 500 kb of each other were aggregated, and an additional 500 kb was added to flank the variants at the extremes within each set. A genomic locus was considered to be novel if all conditionally independent variants within the locus were located ≥250 kb away and not in LD ($R^2$) with any sentinel variant with a $P < 5 \times 10^{-8}$ reported in previously published GWAS of DCM for DCM GWAS or GWAS of any of the three traits included for MTAG in DCM MTAG (Supplementary Table 3).

### Enrichment of Mendelian cardiomyopathy genes within GWAS loci

To estimate the enrichment of Mendelian cardiomyopathy genes within GWAS loci, we first extracted 3,404 genes that had been linked to Mendelian disorder with at least moderate evidence as listed in the ClinGen and GenCC databases (accessed February 2023). We annotated whether each gene was located in GWAS and whether it was listed as one of the 38 Mendelian cardiomyopathy genes (Supplementary Information 4). We then cross-tabulated these annotations and performed statistical tests with one-sided Fisher's exact test to calculate ORs of cardiomyopathy genes being more likely to be situated within GWAS loci. Fisher's exact test was performed using the fisher.test function in R.

### Functionally informed fine-mapping of genomic loci

To prioritize likely causal variants at each genomic locus, we performed functionally informed fine-mapping using PolyFun v.2020-11-14 (ref. 48) and SuSiE v.0.11.92 (ref. 49). Using precomputed prior causal probabilities of 19 million imputed SNPs with MAF > 0.001 based on meta-analysis of 15 traits in UKB from PolyFun, we first estimated per-SNP heritability. These results were then passed to SuSiE to calculate per-SNP posterior inclusion probabilities and to identify 95% credible sets of likely causal variants, assuming at most five causal variants per locus. To run fine-mapping, we used LD reference panels from 10,000 randomly selected UKB European ancestry participants. The procedure was performed separately for loci identified from DCM GWAS and DCM MTAG using the respective summary statistics. For each locus, variants within the identified 95% credible sets in either DCM GWAS or DCM MTAG were aggregated, and annotated with nearest gene(s), genic functions, and Combined Annotation-Dependent Depletion Phred score[50] extracted from ANNOVAR v.2020-06-07 (ref. 51) and OpenTargets Genetics[52].

**Prioritization of effector genes at DCM loci.** To systematically identify and prioritize effector genes at each locus, we followed a two-step approach. First, the nearest gene and the top three genes prioritized by either PoPS[53] or V2G[54] were selected as candidate genes. Second, the totality of evidence including nearest gene, PoPS, V2G and five additional approaches (coding variant, colocalization with gene expression,

TWAS, ABC model, and established Mendelian cardiomyopathy- and muscle-disease-causing genes) was summarized by identifying the number of individual approaches that identified each candidate gene as the most likely, assuming that it met each method's minimum threshold for significance or relevance. Each method received equal weighting, with a maximum score of 8, and the candidate gene with the highest score at each genomic locus was determined to be the prioritized gene. Loci in which gene scores were tied for the highest score were determined not to have a single high-confidence candidate gene.

### Transcriptome-wide association study

We estimated the associations between overall gene expression across tissues and DCM through a multitissue TWAS using eQTL data across 49 human tissues from GTEx v.8 and the DCM GWAS summary statistics implemented in S-MulTiXcan v.0.7.3 with the MASH-R model[55].

### Colocalization with gene expression

To test the hypothesis that genetic associations with gene expression in a given tissue and with DCM are driven by the same causal variants, we performed a statistical colocalization analysis using R coloc v.5.2.3 (ref. 49) allowing for multiple causal variants. The colocalization analysis was performed for all genes overlapping with the identified DCM genetic loci using summary-level eQTL data from GTEx v.8 (ref. 56) in tissues with the lowest TWAS $P$ value and the DCM GWAS summary statistics.

### Polygenic priority score

We computed the polygenic enrichment of gene features derived from cell-type-specific gene expression, biological pathways and protein–protein interactions for all protein-coding genes within the human genome using PoPS v.0.1 (ref. 53). A higher score implies a higher probability of a gene being causal for the trait under study, given feature similarities to other predicted causal genes.

### Variant-to-gene

The V2G model aggregates data from molecular phenotype quantitative trait locus (QTL) experiments including gene expression (eQTL), protein abundance (pQTL) and alternative protein splicing (sQTL), chromatin interaction experiments, in silico functional predictions and genomic distance (between the variant and a gene's canonical transcriptional start site) to compute a variant-level score, with a higher value reflecting greater functional relevance on a given gene[54]. To map variant-level V2G scores onto gene-level scores for gene prioritization, we extracted the V2G score using V2G v.1.1 for all variants that were in LD ($R^2 > 0.8$) with conditionally independent variants or within the fine-mapped variant set for a given locus and took the maximum V2G for a given gene.

### ABC model

The ABC model uses experimental estimates of enhancer activity (assay for transposase-accessible chromatin using sequencing, DNase I hypersensitive site sequencing, or histone 3 K27 acetylation chromatin immunoprecipitation followed by sequencing) and enhancer–promoter contact frequency (high-throughput chromatin conformation capture) to predict enhancer–gene interactions[57]. Precomputed ABC scores generated from experimental data of cardiac left ventricles in ENCODE[58] were identified for the genomic coordinates of fine-mapped and lead variants, with scores >0.02 indicating important interactions.

**Conditional GWAS analysis.** Conditional GWAS analysis was performed using a multitrait-based conditional and joint analysis (mtCOJO) method[59] implemented in GCTA v.1.92.4, which we used to estimate the genetic effects of disease conditioning on AF, CAD, and SBP. To perform the analysis, we used summary statistics from GWAS of AF in 77,690 cases and 1,167,040 controls[60], CAD in 181,522 cases

and 984,168 controls[60] and SBP in 757,601 individuals[61]. For AF and CAD, we calculated the sample prevalence by dividing the number of cases by the number of samples reported in the GWAS, and we used a population prevalence of 2.2% for AF and 7.2% for CAD[62,63]. Given that the vast majority of the GWAS summary statistics used were derived from European ancestry samples, we used 1000G European ancestry to model LD between variants.

**Rare variant gene-based association testing.** Gene-based association testing was performed in the UKB and 100,000 Genomes Project for all genes located within genomic loci, using the genome-wide regression test implemented in REGENIE v.3.2.4. A whole-genome regression model was fitted to allow handling of polygenicity, relatedness and ancestry, using directly genotype-arrayed variants passing quality control (MAF > 0.01, <10% missingness, Hardy–Weinberg equilibrium test $P > 10^{-15}$) in UKB, or directly sequenced variants in the 100,000 Genomes Project (GeL). Next, a gene-based burden test was performed conditional upon the phenotype-specific predictors from the genome-wide regression model and adjusting for sex, age, age[2] and first ten genetic PCs, with body surface area and SBP included as additional covariates for quantitative traits. The outcomes tested were binary case–control status (DCM (narrow and broad definition), heart failure and HCM) and, in the UKB, related CMR quantitative traits (LVESV, LVEDV, LVEF, LV stroke volume and maximum LV wall thickness). Firth correction was applied to account for case–control imbalance. Burden tests collapse variants into a single variable that can be tested for association with a phenotype or trait, thereby reducing computational cost and the test statistic inflation that is seen with other gene-based rare variant tests (for example, SKAT and SKAT-O). Individuals with missing phenotype data were dropped from analysis. For consistency across UKB and GeL, one rare variant mask of PTVs (start lost, stop gained, frameshift, splice acceptor or donor lost) with a MAF $< 1 \times 10^{-3}$ was tested. To minimize the false positive rate resulting from genes with very low allele counts, a minimum allele count (MAC) threshold was applied that considered the approximate sample size: analysis in UKB required MAC $\geq 20$ for binary traits, and MAC $\geq 3$ for quantitative traits; and analysis in GeL required MAC $\geq 3$. A $P$ value FDR-adjusted using the Benjamini–Hochberg method was used for the total number of genes passing the MAC threshold that were tested. Validation of significant associations ($P_{adj} < 0.05$) in any cohort required directional concordance and nominal significance ($P < 0.05$) of the same gene–trait association. Exploratory results evaluating the effect of ultrarare (MAF $< 1 \times 10^{-5}$) variants on binary outcomes in UKB were also tested.

To characterize the overall genetic architecture of DCM, gene-based burden testing of rare PTVs (MAF $< 1 \times 10^{-3}$) was also performed for 16 DCM genes with moderate or definitive evidence[30] in UKB to generate risk estimates for carriers of rare variants with DCM and heart failure.

### Tissue, cell type and cell state heritability enrichment

Tissue-level heritability enrichment analysis was performed using precalculated LD scores of gene expression data from GTEx[56] and chromatin data from the Roadmap Epigenomics[64] and ENCODE[58] projects, with LDSC v.1.0.1 (ref. 65). For cell type and state heritability enrichment, we used the sc-linker[35] approach to link transcriptome-wide gene programs from single-nucleus datasets with GWAS summary statistics. Gene programs derived from snRNA-seq were used to investigate heritability enrichment in cardiac cell types and states using the sc-linker framework[35]. This approach uses snRNA-seq data to generate gene programs that characterize individual cell types and states. These programs are then linked to genomic regions and the SNPs that regulate them by incorporating Roadmap Enhancer-Gene Linking[64,66] and ABC models[57,67]. Finally, the disease informativeness of the resulting SNP annotations is tested using stratified LD score regression,[68] conditional

on broad sets of annotations from the baseline LD model,[41,69] and enrichment statistics and τ coefficients are reported.

Cell-type-specific gene programs were generated from snRNA-seq data of ventricular tissue from 18 control subjects, with cell type annotations made as part of a larger study of ~880,000 nuclei (samples from 52 DCM and 18 control subjects)[36]. Cells that may not have represented true biological states (for example, technical doublets) were excluded from the analysis. For cell type disease-specific programs, pseudobulked counts were used to compare expression levels in DCM and control LV samples within all annotated cell types, using edgeR v.3.32.1 (ref. 70) and methods previously described[36]. Significant DEGs were defined as those with FDR-adjusted $P < 0.05$ and absolute($\log_2$ fold change) > 0.5, requiring a minimum normalized $\log_2$ count of >0.0125 per nucleus (equivalent to 1 count in a nucleus with 10,000 total counts) in either control or DCM samples.

### Pathway enrichment analysis of effector genes, DEGs and intercellular communication in DCM single-nucleus transcriptomics

Pathway gene ontology (GO) enrichment of effector genes and DEGs in DCM was determined at the cell type level and driver GO terms were identified using a two-stage algorithm implemented with gprofiler2 v.0.2.3 (ref. 71). Driver GO terms were determined using a two-stage algorithm implemented with gprofiler2 to identify enriched pathways among GWAS effector genes. GO terms were further examined in the DCM single-nucleus dataset by exploring enrichment among DCM DEGs in all cell types. Functional enrichment analysis was performed using a cumulative hypergeometric probability, with Bonferroni-adjusted P values reported.

To determine the importance of cardiomyocyte and noncardiomyocyte cell types in DCM and the roles of candidate genes and effector-gene-enriched signaling pathways, we explored disease-specific intercellular communication. The single-nucleus transcriptomes of DCM and control samples were interrogated using CellChat v.1.0 for manually curated ligand–receptor interactions (CellChatDB)[37]. In brief, this method identifies overexpressed genes within cell types and states, quantifies the probability of receptor–ligand communication between cells using the law of mass action, and infers statistically and biologically important cellular communications[37]. CellChat was run using default program settings, and the results were analyzed at the cell type level. Endocardial cells were separated from other endothelial cells owing to previously reported important biological effects on ligand–receptor signaling[36]. All analyses were performed in R v.4.0.3.

### Polygenic risk score generation and testing

PGS were generated using a Bayesian framework that models ancestry-specific LD with an external reference set and uses a continuous shrinkage prior, implemented using the PRS-CS v.1.0 package[72]. The phi constant was automatically selected by PRS-CS in an unsupervised approach (PRS-CS auto). Whole-genome PGS scores for all included UKB individuals were calculated using the PLINK 1.9 –score function[73]. Individual SNP weighted scores were generated from DCM GWAS that excluded the UKB cohort, and a subsequent MTAG, to avoid the substantial inflation that occurs when there is overlap of individuals between the GWAS and testing cohorts[74]. The base GWAS summary statistics were filtered to exclude rare and uncommon variants (MAF < 0.01) and ambiguous SNPs that were not resolvable by strand-flipping. We calculated a PGS for unrelated (third degree or closer) White British participants in the UKB (application number 47602) using variants that passed genotyping quality control (MAF > 0.01, genotyping rate >0.99, Hardy–Weinberg equilibrium test $P > 1 \times 10^{-6}$). Variants overlapping the base, target and LD reference set (1000 Genomes Project phase 3 European ancestry) were included. PGS predictive performance was assessed on the basis of

AUROC and association with DCM and associated CMR traits (OR per PGS standard deviation and comparing top quantiles with the median) in the UKB, and in carriers of rare variants predicted to cause DCM[30] (see Supplementary Information 5 for full details of variant curation and genes tested). All models included age, $age^2$, sex and first ten genetic PCs as covariates. AUROC was calculated for logistic regression models using pROC v.1.18.4, randomly separating the cohort into 70% generation and 30% evaluation. Nagelkerke's $R^2$ was calculated using fmsb v.0.7.5 with the null model only including age, $age^2$, sex and first ten genetic PCs as covariates. Time-to-event analysis was performed using survival v.3.5.7, and cumulative incidence curves were generated using survminer v.0.4.9. All statistical analyses were performed in R v.4.0.3.

### Phenome-wide association study

The pleiotropic effects of genetic risk arising from common variants were tested by performing a pheWAS of PGS in the UKB. ICD-9 and ICD-10 codes from death records and hospital admission episodes were translated to Phecodes (Phecode Map 1.2)[75]. For binary phenotypes with at least 20 cases, PGS–phenotype association was tested using logistic regression adjusted for age, $age^2$, sex and first ten genetic PCs as covariates. Sensitivity analyses adjusting for DCM or heart failure and hypertension status in the regression model were performed to identify independent effects. The significance threshold was adjusted for the total number of phenotypes tested ($P < 2.72 \times 10^{-5}$), and data were presented using Manhattan plots, grouped by body system. PheWAS were performed using PheWAS v.2018-03-12 (ref. 76) in R v.4.0.3.

### Reporting summary

Further information on research design is available in the Nature Portfolio Reporting Summary linked to this article.

## Data availability

Data from UKB can be requested from the UKB Access Management System (https://www.ukbiobank.ac.uk/enable-your-research/apply-for-access). Data from the 100,000 Genomes Project can be accessed following an application to join the Genomics England Clinical Interpretation Partnership (https://www.genomicsengland.co.uk/research/academic/join-research-network). The ClinGen (https://www.clinicalgenome.org) and GenCC (https://search.thegencc.org) databases can be directly accessed. GWAS summary statistics are available on the Cardiovascular Disease Knowledge Portal (https://cvd.hugeamp.org/dinspector.html?dataset=Zheng2024_DCM_EU). Regional association plots for all 80 risk loci are available online (https://hermes-dcm-locus.netlify.app). The PGS are available for download at the Polygenic Score Catalog (https://www.pgscatalog.org/) under accession IDs PGS004861 and PGS004862. The raw single-nucleus gene expression dataset is available for download from the European Phenome-Genome Archive (dataset ID EGAD00001009292).

## Code availability

Custom analysis code to perform the main GWAS analyses is available via Zenodo at https://doi.org/10.5281/zenodo.11204854 (ref. 77). Additional analyses were performed using publicly available software as described in the Methods section.

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

## Acknowledgements

We acknowledge contributions from the 100,000 Genomes Project, COVIDsortium, DBDS Genomic Consortium, Estonian Biobank and HERMES Consortium. This work was supported by funding from the British Heart Foundation (RE/18/4/34215, FS/IPBSRF/22/27059, FS/15/81/31817, FS/ICRF/21/26019, RG/19/6/34387, BC/F/21/220106, FS/18/65/34186, SP/19/1/34461, SP/17/11/32885, CH/P/23/80008, RE/24/130023), the Medical Research Council (MC_UP_1605/13), Wellcome Trust (107469/Z/15/Z); National Institute for Health Research (NIHR) Imperial College Biomedical Research Centre, NIHR Royal Brompton Cardiovascular Biomedical Research Unit, Sir Jules Thorn Charitable Trust (21JTA), National Heart and Lung Foundation, Royston Centre for Cardiomyopathy Research, Rosetrees Trust, GenMED LABEX, UCL British Heart Foundation Research Accelerator and NIHR University College London Biomedical Research Centre. This research was conducted in part using the UKB resource under application numbers 9922, 15422, 18545, 40616 and 47602 and was made possible through access to data in the National Genomic Research Library, which is managed by Genomics England Limited (a wholly owned company of the Department of Health and Social Care). The National Genomic Research Library holds data provided by patients and collected by the NHS as part of their care and data collected as part of their participation in research. The National Genomic Research Library is funded by the NIHR and NHS England; the Wellcome Trust, Cancer Research UK and the Medical Research Council have also funded research infrastructure. Individual study acknowledgements are reported in Supplementary Information 6. The views expressed

## Author contributions

S.L.Z. and A.H. conceived, designed and performed the experiments, performed statistical analysis, analyzed the data, wrote the paper with input from all authors and contributed equally to this work. J.S.W. and R.T.L. conceived and designed the experiments, contributed data, wrote the paper and jointly supervised this work. M.L., K.M., X.X. and C.F. performed statistical analysis and analyzed the data. D.C., D.M., I.B., H.I., A.d.M., P.I., R.B., D.S., E.A., L.J.A., K.G.A., J.A., J.B., A.J.B., P.J.R.B., K.J.B., E.B., J.B., S.B., H.B., D.J.C., P.C., J.P.C., S.A.C., S.D., J.-F.D., A.D., P.E., T.E., C.E., E.H.F.-F., C.F., S.G., J.G., V.G., D.G., C.M.H., B.P.H., A.H., H. Hemingway, H.L.H., L.L., C.M.L., B.D.L., K.M., I.R.M., M.P.M., A.D.M., A.P.M., L.M., C.M., J.C.M., M. Noursadeghi, A.T.O., S.R.O., C.N.A.P., A.P., S.K.P., O.B.P., A.A.R., A.S., D.T.S., S.S., K.S., G.S., P.S., M.L.-T., U.T., T.A.T., M.T.-L., G.T., U.T., V.T., D.-A.T., H.U., A.M.V., J.v.S., M.v.V., A.V., M.V., E.V., COVIDsortium, DBDS Consortium, HERMES Consortium and Genomics England Research Consortium contributed data. T.P.C., M.-P.D., M.D., P.T.E., A.D.H., C.C.L., N.J.S., S.H.S., J.G.S., R.S.V., D.P.O.'R., H. Holm, M. Noseda and Q.S.W. conceived and designed experiments and contributed data.

## Competing interests

S.L.Z. has acted as a consultant for Health Lumen. A.H. and R.T.L. have received funding from Pfizer Inc. R.T.L. has performed paid consultancy for Health Lumen and Fitfile Ltd. J.S.W. has acted as a consultant for MyoKardia, Pfizer, Foresite Labs and Health Lumen and received institutional support from Bristol Myers Squibb and Pfizer Inc. P.C. has received personal fees for consultancies, outside the present work, for Amicus, Pfizer Inc., Owkin and Bristol Myers Squibb. M.-P.D. declares holding equity in Dalcor Pharmaceuticals, unrelated to this work. The authors who are affiliated with deCODE genetics/Amgen Inc. and Regeneron Pharmaceuticals declare competing financial interests as employees. The other authors declare no competing interests.

## Additional information

**Extended data** is available for this paper at https://doi.org/10.1038/s41588-024-01952-y.

**Correspondence and requests for materials** should be addressed to James S. Ware or R. Thomas Lumbers.

v

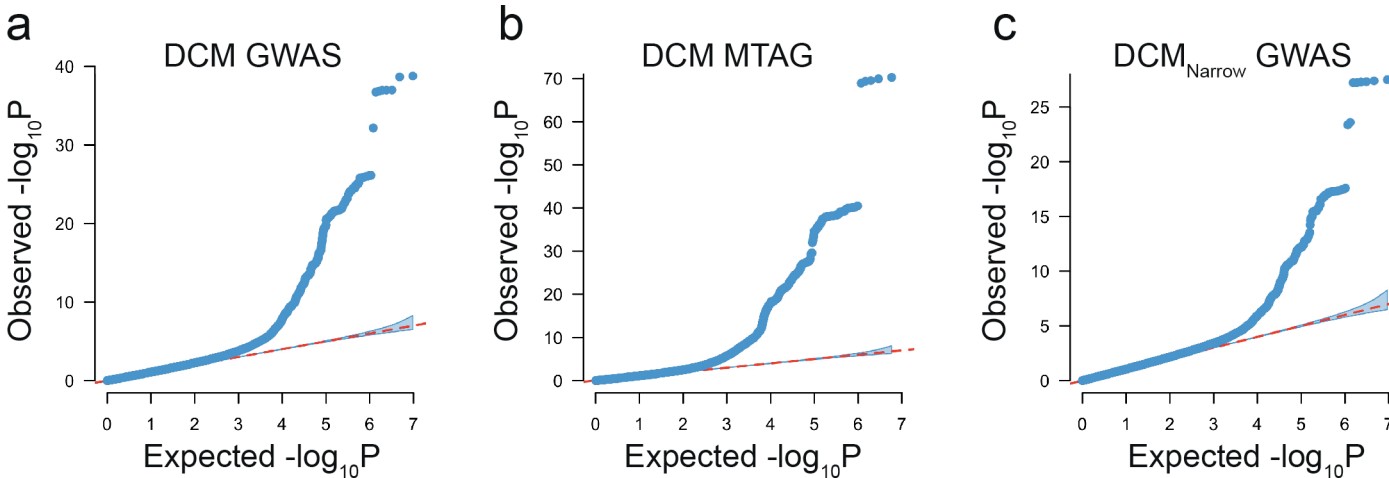

**Extended Data Fig. 1 | Quantile-quantile plots.** Quantile-quantile plots for (**a**) DCM GWAS, (**b**) DCM MTAG and (**c**) DCM$_{Narrow}$ GWAS. The shaded error bar indicates the 95% confidence interval under the assumption of a uniform distribution of P values (red dashed line).

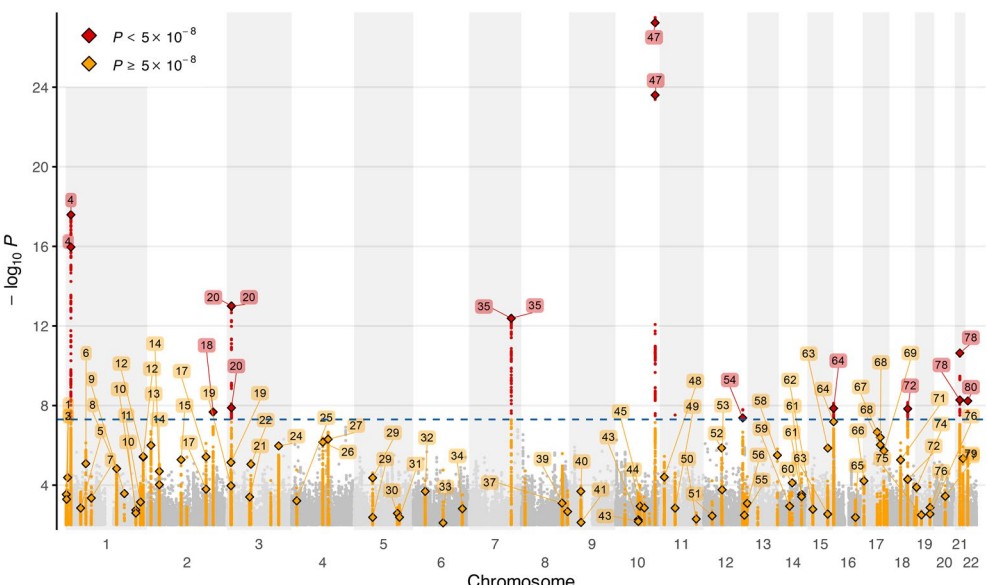

**Extended Data Fig. 2 | Manhattan plot of DCM$_{Narrow}$ GWAS.** Manhattan plot of GWAS of 6,001 strictly defined DCM cases and 449,384 controls (DCM$_{Narrow}$ GWAS). GWAS was performed using the same methods as for DCM GWAS using the subset of studies that recruited participants from specialist clinical cohorts or using unequivocal DCM diagnostic codes (Supplementary Information 1). DCM diagnosis required cardiac imaging, clinical expertise and/or robustly-defined ICD codes. The 80 loci identified from DCM GWAS and DCM MTAG (Fig. 2) are labelled. In total there were 10 loci reaching genome-wide significance (dashed blue line – P < 5 × 10$^{-8}$), all of which were significant in the primary GWAS. P-values were two-sided and based on inverse-variance weighted fixed-effects model, and not adjusted for multiple testing.

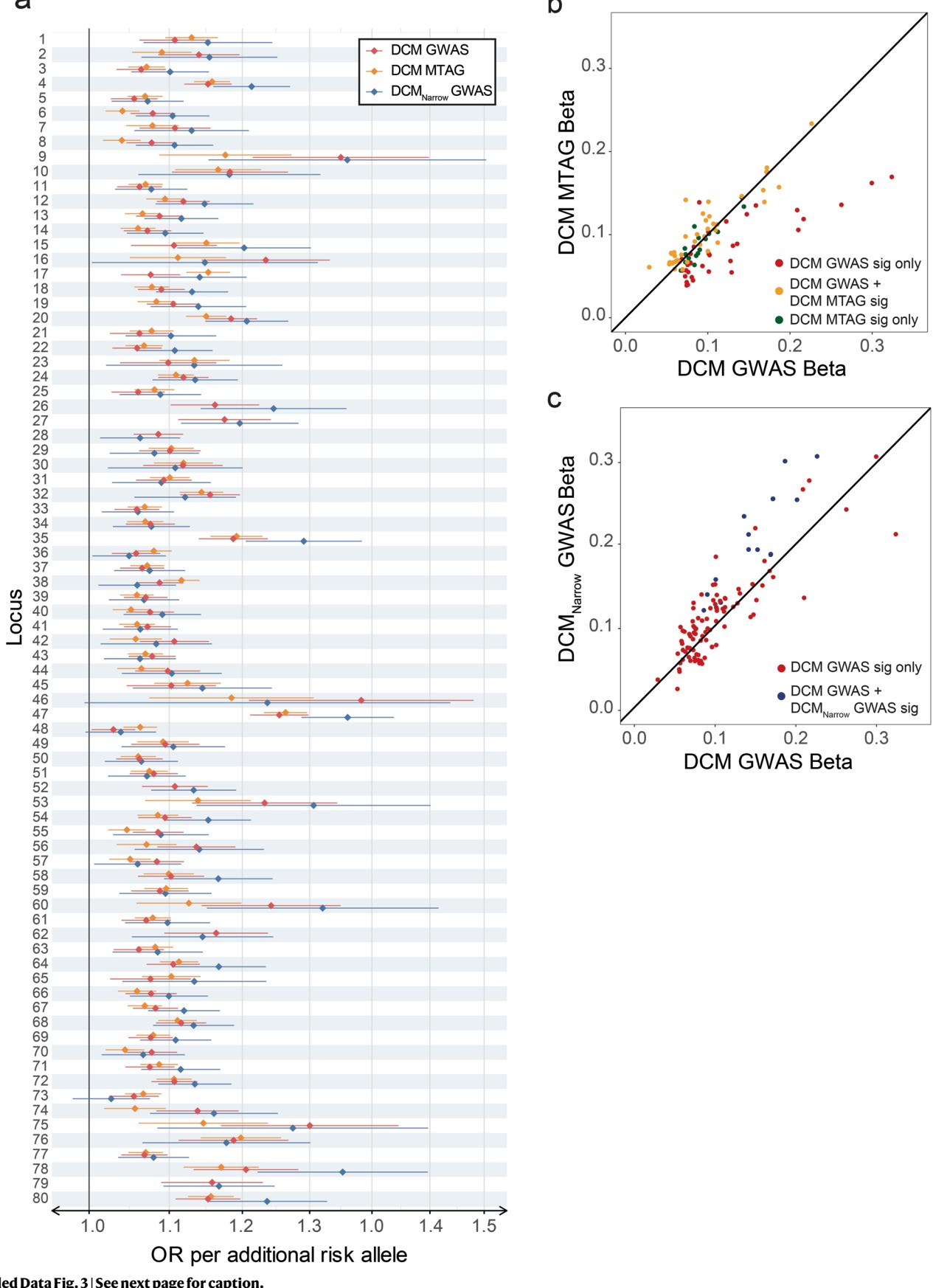

**Extended Data Fig. 3 | See next page for caption.**

**Extended Data Fig. 3 | Comparison of effect sizes across DCM GWAS, DCM MTAG, and DCM$_{Narrow}$ GWAS. a,** Forest plot of effect size across DCM GWAS, DCM MTAG and DCM$_{Narrow}$ GWAS for all 80 genomic risk loci identified in DCM GWAS and DCM MTAG. Effect estimates are derived from DCM GWAS of 12,556 cases and 1,199,156 controls (red), DCM MTAG consisting of the DCM GWAS cohort and 36,203 participants with cardiac magnetic resonance derived quantitative cardiac traits (orange), and DCM$_{Narrow}$ GWAS of 6,001 cases and 449,382 controls (blue). All sentinel variants at the 80 genomic risk loci identified in this study are presented (62 from DCM GWAS using FDR threshold 1% and 54 from DCM MTAG at genome-wide significance). The central effect estimate is represented with a diamond and the tails represent the 95% confidence interval.

**b,** Scatter plot comparing absolute effect sizes for conditionally independent variants in DCM GWAS and DCM$_{Narrow}$ GWAS. **c,** Scatter plot comparing absolute effect sizes for conditionally independent variants in DCM GWAS and DCM MTAG. Variants tended to have a greater effect in DCM$_{Narrow}$ GWAS than in DCM GWAS, particularly for variants that were genome-wide significant in DCM$_{Narrow}$ GWAS (**blue**) compared with those that were only FDR significant in DCM GWAS (**red**). When comparing DCM GWAS and DCM MTAG, variants that were FDR significant in DCM GWAS and genome-wide significant in DCM MTAG (**dark green**), and that were genome-wide significant only in DCM MTAG (**yellow**), had similar effect sizes, while variants that were only FDR significant in DCM GWAS (**red**) tended to have larger effects in DCM GWAS than in DCM MTAG.

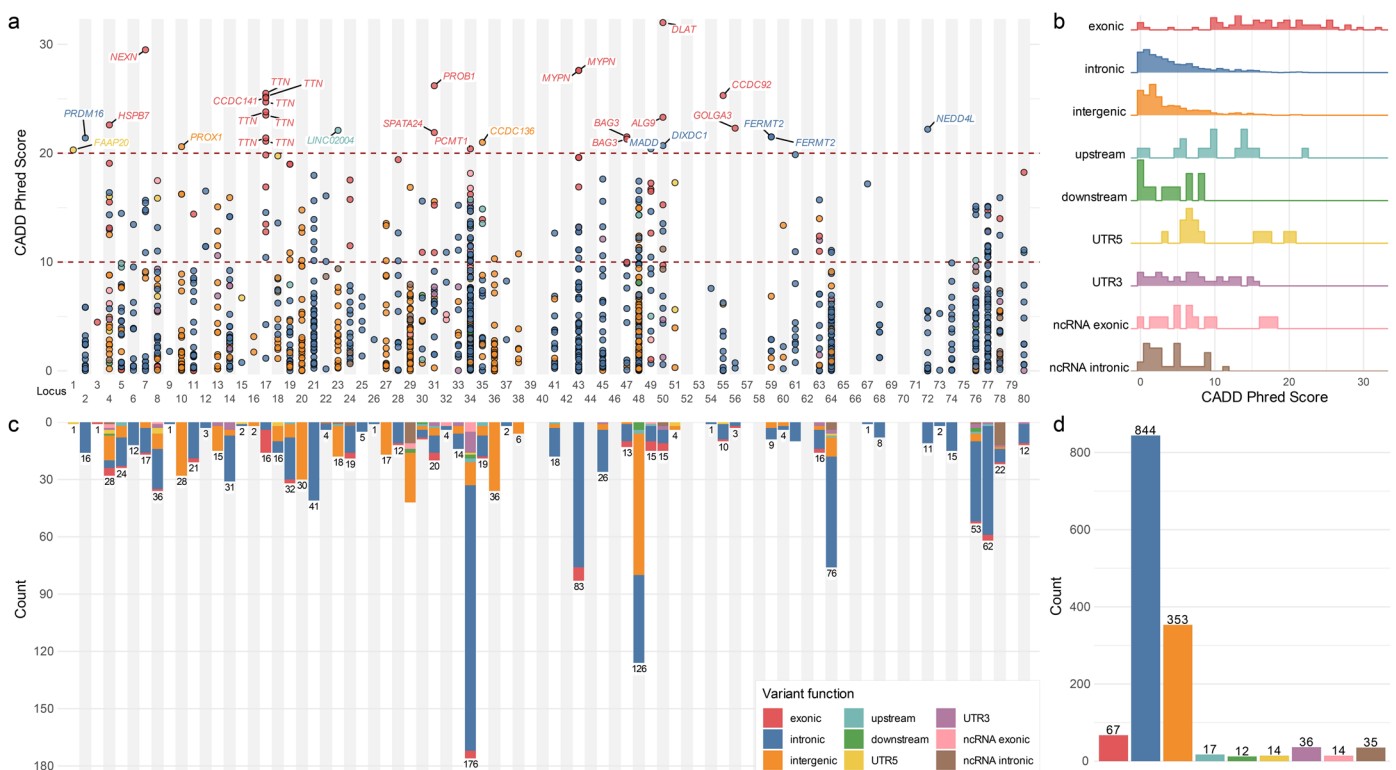

**Extended Data Fig. 4 | Functionally-informed fine-mapped variants at genomic loci. a**, Fine-mapped variants at genomic risk loci with variants with high CADD Phred scores (>20) annotated to the nearest gene. **b**, Total number and function of fine-mapped variants at each locus. **c**, Distribution of CADD Phred scores for fine-mapped variants across all genomic risk loci, stratified by variant function. **d**, Number of fine-mapped variants stratified by function.

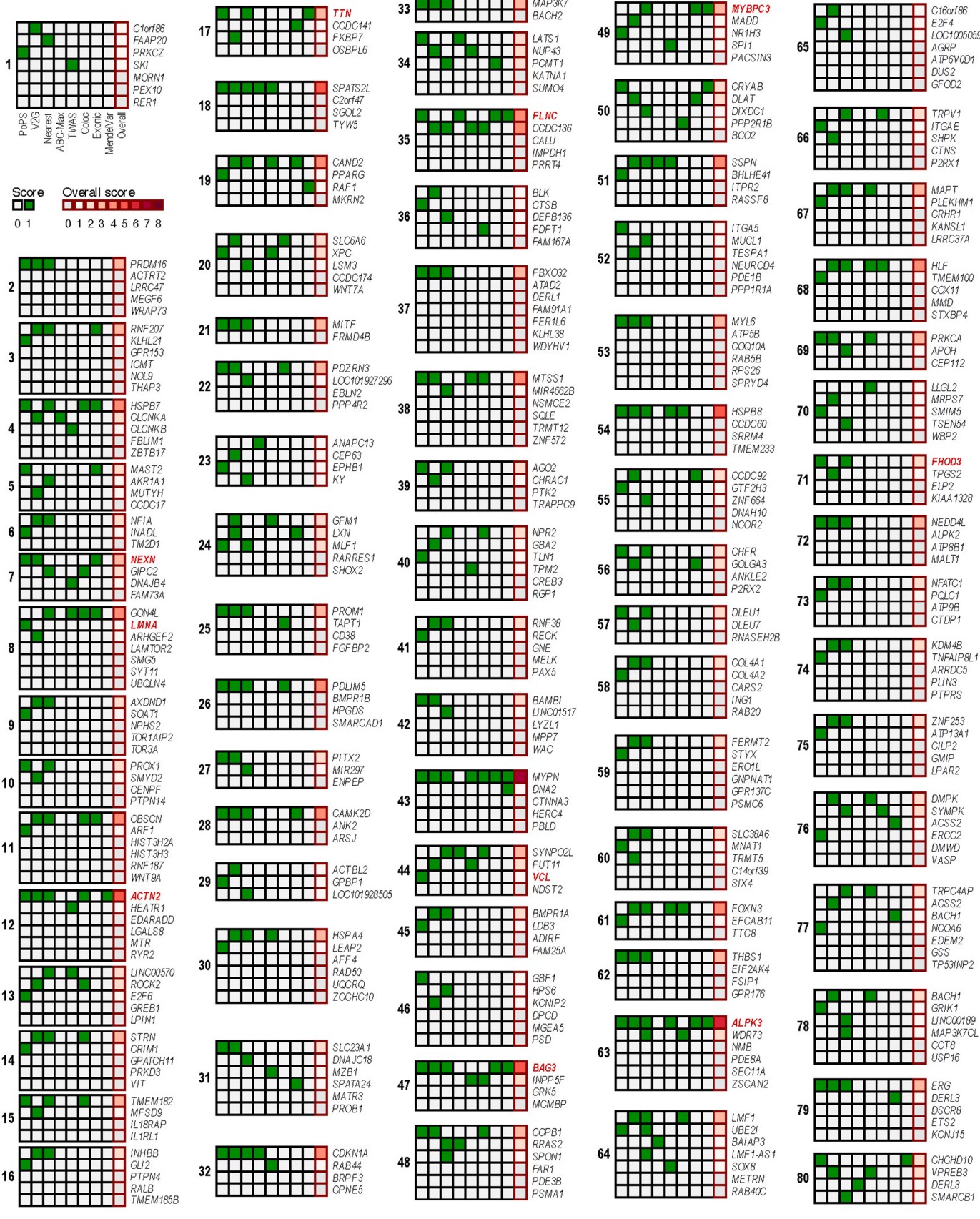

**Extended Data Fig. 5 | See next page for caption.**

**Extended Data Fig. 5 | Summary of effector gene prioritization results.**
A two-step approach was used to identify candidate genes and prioritize potential effector genes at each loci. First, the nearest gene along with the top 3 genes scored using each of PoPs and V2G were highlighted as candidate genes for further evaluation. Second, of these genes, 5 additional features and methods were used to score the overall level of evidence supporting each putative gene by giving one point for any gene that was identified as best from each feature (maximum score of 8), and the highest scoring gene(s) at each locus being identified as the candidate gene(s). The 8 features were: PoPs, V2G, nearest, activity-by-contact (ABC)-model, transcriptome-wide association study (TWAS), colocalization, exonic coding variant, and reported Mendelian cause of cardiomyopathy or muscle disorder. Highlighted in red are genes with moderate or definitive evidence of being Mendelian causes of cardiomyopathy from ClinGen curation.

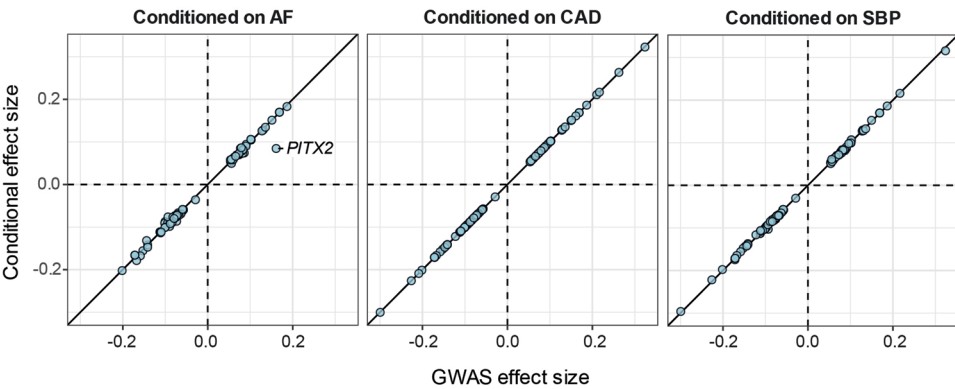

**Extended Data Fig. 6 | Conditional analysis of GWAS on atrial fibrillation, coronary artery disease, and systolic blood pressure.** Comparison of effect estimates from the original DCM GWAS (X axis) and from conditional GWAS on atrial fibrillation (AF), coronary artery disease (CAD), and systolic blood pressure (SBP) (Y-axis).

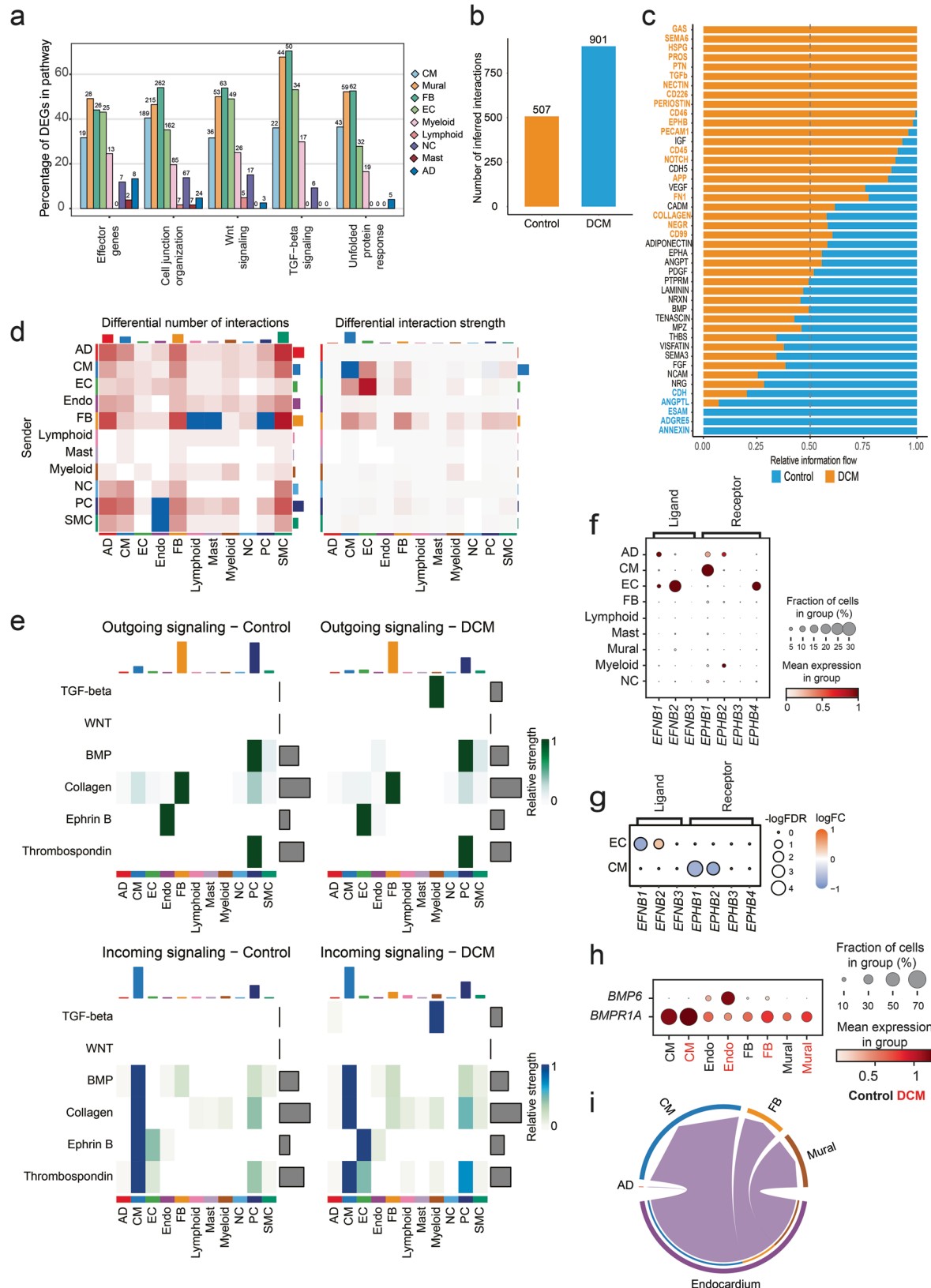

**Extended Data Fig. 7 | See next page for caption.**

**Extended Data Fig. 7 | Intercellular interactions in DCM inferred from single nuclei transcriptomics. a**, Percentage of genes within candidate gene enriched pathways that are differentially expressed in DCM compared with controls, stratified by cell type. **b**, Total number of interactions between cell types in DCM (blue) and control (orange). **c**, Relative information flow of curated receptor-ligand intercellular, highlighting pathways that are significantly increased in DCM (orange) or control (blue). **d**, Heat map showing total overall differences in interaction number and strength between cell types (red – increased in DCM, blue – decreased). **e**, Heat map showing outgoing (green) and incoming (blue) signals for prioritized gene enriched pathways (TGF-beta and WNT pathways) and specific pathways of prioritised genes (BMP, Collagen, Ephrin B and thrombospondin). **f**, Expression levels of ephrin-B ligand and receptors across major cell types. Mean expression is scaled from minimum to maximum, and proportion of expressing nuclei within a cell type indicated by dot size.

**g**, Increased expression of *EFNB2* (ligand) in endothelial cells (EC) and decreased expression of *EPHB1* (receptor) in cardiomyocytes (CM) in DCM. Dot colour represents change in expression compared with control, and dot size represents the FDR-adjusted *P*-value. **h**, Expression levels of *BMP6* and *BMPR1A* in CM, endocardial, fibroblast (FB), and mural nuclei, stratified by HCM (red) and control (black) status. Mean expression is scaled from minimum to maximum, and proportion of expressing nuclei within a cell type indicated by dot size. **i**, Chord plot showing that majority of endocardial (purple) BMP6-BMPR1A signaling is to cardiomyocytes (blue), followed by mural (brown) and fibroblasts (orange). Dot colour reflects the communication probabilities and dot size represents *P*-values computed from one-sided permutation test. AD – adipocyte; CM – cardiomyocyte; EC – endothelial cell; Endo – endocardial cell; FB – fibroblast; NC – neuronal cell; PC – pericyte; and SMC – smooth muscle cell.

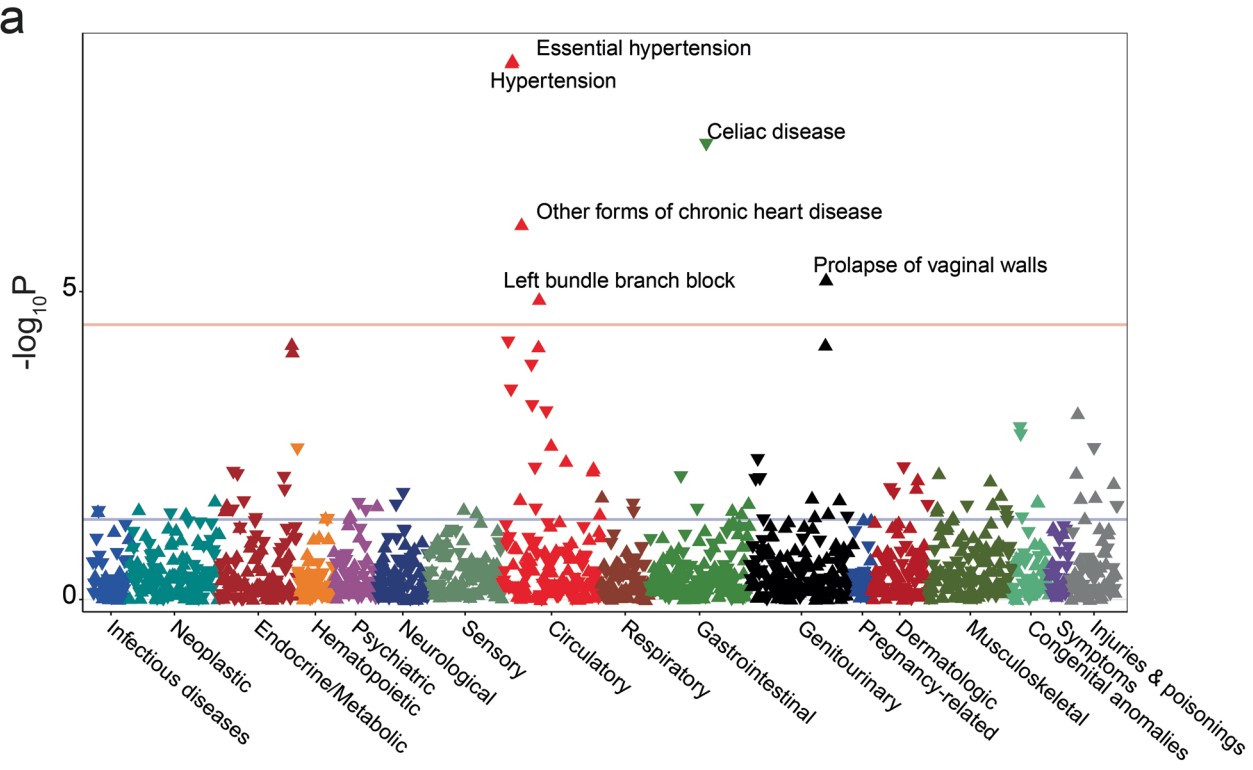

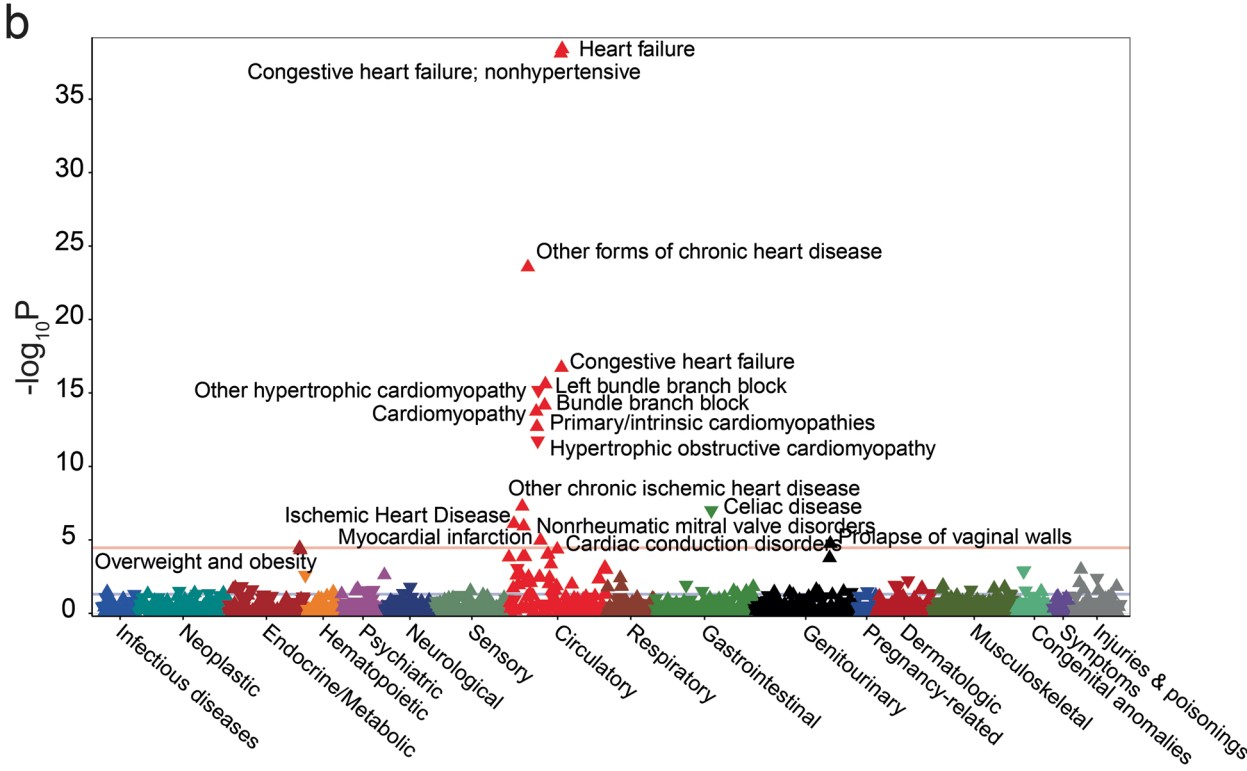

**Extended Data Fig. 8 | DCM-PGS pheWAS adjusted for DCM/heart failure, and hypertension.** Manhattan plot of DCM-PGS associations after adjusting for DCM or heart failure (**a**), and hypertension (**b**) status in UK Biobank. Additional co-variates included in the linear regression model include sex, age, age², and first ten principal components. ICD-9 and ICD-10 diagnostic codes are mapped to Phecode Map version 1.2. Mapped phenotypes exceeding phenome-wide significance threshold (P 2.7 ×10⁻⁵, red line, adjusted for the total number of tested phenotypes) are labelled. Blue line indicates nominal significance (P < 0.05). Direction of triangle indicates the direction of effect of the PGS association. P-values are two-sided and calculated from linear regression model, and not adjusted for multiple testing.

# natureportfolio

# Reporting Summary

## Statistics

For all statistical analyses, confirm that the following items are present in the figure legend, table legend, main text, or Methods section.

| n/a | Confirmed | |
|---|---|---|
| ☐ | ☒ | The exact sample size (*n*) for each experimental group/condition, given as a discrete number and unit of measurement |
| ☐ | ☒ | A statement on whether measurements were taken from distinct samples or whether the same sample was measured repeatedly |
| ☐ | ☒ | The statistical test(s) used AND whether they are one- or two-sided *Only common tests should be described solely by name; describe more complex techniques in the Methods section.* |
| ☐ | ☒ | A description of all covariates tested |
| ☐ | ☒ | A description of any assumptions or corrections, such as tests of normality and adjustment for multiple comparisons |
| ☐ | ☒ | A full description of the statistical parameters including central tendency (e.g. means) or other basic estimates (e.g. regression coefficient) AND variation (e.g. standard deviation) or associated estimates of uncertainty (e.g. confidence intervals) |
| ☐ | ☒ | For null hypothesis testing, the test statistic (e.g. *F*, *t*, *r*) with confidence intervals, effect sizes, degrees of freedom and *P* value noted *Give P values as exact values whenever suitable.* |
| ☒ | ☐ | For Bayesian analysis, information on the choice of priors and Markov chain Monte Carlo settings |
| ☒ | ☐ | For hierarchical and complex designs, identification of the appropriate level for tests and full reporting of outcomes |
| ☐ | ☒ | Estimates of effect sizes (e.g. Cohen's *d*, Pearson's *r*), indicating how they were calculated |

*Our web collection on statistics for biologists contains articles on many of the points above.*

## Software and code

Policy information about availability of computer code

| Data collection | No software was used for data collection. |
|---|---|
| Data analysis | GWAS meta-analysis was performed centrally using METAL v2020-05-05. SNP-based heritability was calculated using LDAK SumHer software v5.2. Genetic correlation between traits was performed using LDSC v1.0.1. Multitrait analysis of GWAS was performed using mtag v1.0.8. Conditionally independent variants were identified using GCTA software v1.92.4. Prioritisation of causal variants was performed using fine-mapping with PolyFun v2020-11-14 and SuSiE v0.11.92. Polygenic prediction score (PoPS) v0.1, OpenTargets Variant2Gene v1.1, S-MultiXcan v0.7.3, and coloc R package  v5.2.3 were used for effector gene prioritisation. Rare variant burden testing was performed using Regenie v3.2.4. Pathway enrichment of prioritised genes was performed using g:profiler v0.2.3 package in R. Intercellular communication was performed using CellChat v1.0 package in R, and differential gene expression performed using edgeR v3.32.1 package in R. Tissue-based and cell-type heritability estimation was performed using S-LDSC v1.0.1. Polygenic scores were derived using PRS-CS auto v1.0, with individual PGS scores generated using PLINK v1.9. Phenome-wide association study of PGS was performed using phewas v2018-03-12 package in R. |

For manuscripts utilizing custom algorithms or software that are central to the research but not yet described in published literature, software must be made available to editors and reviewers. We strongly encourage code deposition in a community repository (e.g. GitHub). See the Nature Portfolio guidelines for submitting code & software for further information.

## Data

Policy information about availability of data

All manuscripts must include a data availability statement. This statement should provide the following information, where applicable:

- Accession codes, unique identifiers, or web links for publicly available datasets
- A description of any restrictions on data availability
- For clinical datasets or third party data, please ensure that the statement adheres to our policy

Data from UK Biobank can be requested from the UK Biobank Access Management System. Data from 100,000 Genomes Project can be accessed following application to join the Genomics England Clinical Interpretation Partnership. ClinGen databases are accessible at https://www.clinicalgenome.org and GenCC databases are accessible at https://search.thegencc.org. GWAS summary statistics are available to download from the Cardiovascular Disease Knowledge Portal (https://api.kpndataregistry.org/api/d/Hv5oWo). Regional association plots for all 80 risk loci are available online (https://hermes-dcm-locus.netlify.app). The PGS are available for download on the Polygenic Score Catalog (https://www.pgscatalog.org/) under accession IDs PGS004861 and PGS004862. The raw single nuclei gene expression dataset is available for download from the European Phenome-Genome Archive (Dataset ID EGAD00001009292).

## Human research participants

Policy information about studies involving human research participants and Sex and Gender in Research.

| | |
|---|---|
| Reporting on sex and gender | The article uses the term sex when referring to biological attribute, and was determined using genetic sex where available. Sex was included as a covariate in all multivariate analyses. Findings are relevant to both male and females. |
| Population characteristics | Population characteristics include age, sex, ancestry (self-reported and genetic) and genetic principal components for all individuals. Blood pressure and body surface area was available for individuals in the cardiac magnetic resonance imaging substudy of the UK Biobank. Baseline characteristics for cohorts are reported in Supplementary Table 1. |
| Recruitment | Participants were recruited to the UK Biobank from a large number of national sources (e.g. GP, leaflets and advertising, hospitals, and recruitment drives in the community), and targeted individuals from middle age onwards. 100,000 Genomes Project recruited patients with rare disease and cancer along with their relatives, from clinical centres, initially with an emphasis on genetically unexplained disease. Individual study details on participant recruitment is provided in the Supplementary Methods. |
| Ethics oversight | All patients gave written informed consent, and all studies were approved by the relevant regional research ethics committees, and adhered to the principles set out in the Declaration of Helsinki. The UK Biobank study was reviewed by the National Research Ethics Service (11/NW/0382, 21/NW/0157). The 100,000 Genomes Project was reviewed by the National Research Ethics Service (14/EE/1112 and 13/EE/032). |

Note that full information on the approval of the study protocol must also be provided in the manuscript.

# Field-specific reporting

Please select the one below that is the best fit for your research. If you are not sure, read the appropriate sections before making your selection.

☒ Life sciences ☐ Behavioural & social sciences ☐ Ecological, evolutionary & environmental sciences

For a reference copy of the document with all sections, see nature.com/documents/nr-reporting-summary-flat.pdf

# Life sciences study design

All studies must disclose on these points even when the disclosure is negative.

| | |
|---|---|
| Sample size | No sample size calculations were made. We used the maximum number of available cases and controls that passed quality control thresholds/metrics. |
| Data exclusions | No data were excluded. |
| Replication | No replication of identified genetic risk loci was performed. This was done to maximise the discovery set. The findings are therefore reported as exploratory, albeit in the context of large sample size numbers. Previous studies from this current group of authors that have used MTAG (Tadros et al. Nat Genet 2020) have subsequently shown in larger sample sizes that replication of loci occurs (Tadros, Zheng et al. Nat Genet under review). Furthermore, there are no additional suitable studies that the authors are aware of to conduct replication in. |
| Randomization | Observational study - not applicable |
| Blinding | Observational study - not applicable |

# Reporting for specific materials, systems and methods

We require information from authors about some types of materials, experimental systems and methods used in many studies. Here, indicate whether each material, system or method listed is relevant to your study. If you are not sure if a list item applies to your research, read the appropriate section before selecting a response.

## Materials & experimental systems

| n/a | Involved in the study |
|-----|----------------------|
| ☒ ☐ | Antibodies |
| ☒ ☐ | Eukaryotic cell lines |
| ☒ ☐ | Palaeontology and archaeology |
| ☒ ☐ | Animals and other organisms |
| ☒ ☐ | Clinical data |
| ☒ ☐ | Dual use research of concern |

## Methods

| n/a | Involved in the study |
|-----|----------------------|
| ☒ ☐ | ChIP-seq |
| ☒ ☐ | Flow cytometry |
| ☒ ☐ | MRI-based neuroimaging |

