## [Peer Review File · Nature Genetics]

Peer Review Information

Manuscript Title: Genome-wide association analysis provides insights into the molecular etiology of dilated cardiomyopathy

Corresponding author name(s): Dr R (Thomas) Lumbers, Professor James (S) Ware

Reviewer Comments & Decisions:

Decision Letter, initial version:

23rd Oct 2023

Dear Dr Lumbers,

Your Letter, "Genome-wide association analysis reveals insights into the molecular aetiology underlying dilated cardiomyopathy" has now been seen by 2 referees. You will see from their comments below that while they find your work of interest, some important points are raised. We are interested in the possibility of publishing your study in Nature Genetics, but would like to consider your response to these concerns in the form of a revised manuscript before we make a final decision on publication.

To guide the scope of the revisions, the editors discuss the referee reports in detail within the team with a view to identifying key priorities that should be addressed in revision. In this case, we think both referees have identified important aspects of the presentation and interpretation of the results that need to be improved. Reviewer #1 has a major concern regarding the threshold used for genome-wide significance ($FDR < 0.01$), and suggests using a more stringent ($P < 5 \times 10^{-8}$) threshold; another concern is about specificity of some associations. Reviewer #2 notes that the statement of "66 novel associations with DCM" needs further clarification or additional support; and the description about primary GWAS as "DCM broad" might be misleading. We particularly ask that you address their technical comments as thoroughly as possible with additional analyses and appropriate revisions. We hope that you will find the prioritized set of referee points to be useful when revising your study.

We therefore invite you to revise your manuscript taking into account all reviewer and editor comments. Please highlight all changes in the manuscript text file. At this stage we will need you to upload a copy of the manuscript in MS Word .docx or similar editable format.

*1) Include a “Response to referees” document detailing, point-by-point, how you addressed each referee comment. If no action was taken to address a point, you must provide a compelling argument. This response will be sent back to the referees along with the revised manuscript.

*2) If you have not done so already please begin to revise your manuscript so that it conforms to our Letter format instructions, available here.

*3) Include a revised version of any required Reporting Summary:

Please be aware of our guidelines on digital image standards.

[redacted]

We hope to receive your revised manuscript within 3 to 6 months. If you cannot send it within this time, please let us know.

Sincerely,
Wei

Wei Li, PhD
Senior Editor
Nature Genetics
New York, NY 10004, USA
www.nature.com/ng

Reviewers' Comments:

Reviewer #1:

Remarks to the Author:

This paper by Lumbers, Ware and colleagues reports a large GWAS meta-analysis of dilated cardiomyopathy including 14K cases and > 1M controls. The authors report a large number of loci and complement their effort with a rich set of ancillary analyses. The paper will contribute significantly to the field and is of overall good quality. I do have some concerns:

1) The authors report novel loci according to a relatively liberal FDR threshold. How many loci are novel using a stricter Bonferroni correction? This would be preferable given the risk for false-positive results. Figure S5 suggests most reported associations are in fact not quite genome-wide significant ($P < 5 \times 10^{-8}$).

Also, it's not entirely fair to previous publications who used stricter criteria to define genome-wide significance.

2) The MTAG analysis is interesting, but there is a risk some variants might be associated with cardiac MRI traits yet are not relevant to DCM per se. This is similar to the observation many loci associated with fasting glucose are not relevant to risk of diabetes. Can the authors comment on this?

3) The authors prioritize PITX2, a well-known atrial fibrillation gene. As atrial fibrillation is itself a risk factor for heart failure, can the authors exclude the possibility some of the genes are primarily heart rhythm genes rather than DCM ones? Is there evidence PITX2 can impact DCM risk independently of AF? Importantly, DCM itself can also lead to atrial fibrillation so the relationship could well be bi-directional (i.e. DCM "unmasking" atrial fibrillation).

4) The PRS phenome-wide analysis also suggests lack of specificity. How many of the prioritized genes are specific to DCM versus primarily associated with a risk factor for DCM (such as atrial fibrillation, CAD, etc.)? Loci primarily associated with DCM risk factors should be excluded from ancillary analyses unless there is a good rationale not to do so.

5) How does the PGS compares to carrying a bona fide DCM Mendelian mutation in terms of DCM risk?

6) Paragraph starting at line 291: What do the authors refer to by "P_{adj}"? How was the p-value adjusted. More generally, lots of intriguing results are presented yet it is not clear which ones are significant beyond multiple testing adjustment. It is also not clear how many hypotheses were tested in the first place, presumably a large number.

Minor

7) "The conversion to liability scale was calculated using a population prevalence of 0.004 for DCM-Strict (based on estimated prevalence of 1 in 250 individuals) and 0.08 for DCM (assuming twice the prevalence of DCM-Strict)." Did the authors use 0.008 or 0.08?

8) Figure 3-C: Can the authors explain the what bolded and "*" genes represent in the figure legend?

9) Figure 4: "The grey highlighted region indicates smoothed regression lines of the upper and lower bounds for each effect estimate." As loci were selected to have nominal association ($p < 0.05$), it is very difficult to interpret. Even under the null of no association, a similar pattern would be observed. Limiting to genes significant after adjustment for multiple hypothesis testing would be useful.

Reviewer #2:

Remarks to the Author:

Thank you for the opportunity to review the manuscript, "Genome-wide association analysis reveals insights into the molecular etiology underlying dilated cardiomyopathy," by Zheng and colleagues. This manuscript provides extensive human genetic analyses of non-ischemic cardiomyopathy (NICM), with a particular interest in understanding the genetics of dilated cardiomyopathy (DCM). The authors perform common variant genome-wide association study (GWAS) meta-analyses and multi-trait of GWAS (MTAG) analyses, followed by typical downstream bioinformatics to identify candidate causal SNPs and genes. They further use sequencing data to examine rare variants in a set of candidate genes, and finally, they assess the relevance of polygenic versus monogenic risk, corroborating prior studies that have shown that polygenic risk has modifying effect on Mendelian disease. A major limit in studying the genetics of NICM to date has been sample size, and efforts like this one that bring together and harmonize multiple datasets are important. Overall, I find this manuscript to be well written. The methods are generally sound, and the results are interesting. My main critiques are below.

MAJOR:

1. The authors describe their primary GWAS as "DCM broad." I think this description and their subsequent interpretation may be a little misleading. Although this critique may reflect semantics, precise terminology remains quite important in clinical settings. The presence or absence of left ventricular dilation is a central diagnostics and prognostic factor. The phenotype of LV systolic dysfunction without LV dilation could represent early stages of DCM, but this phenotype is not well accepted as a category of DCM. Indeed, the 2023 European Society of Cardiology guidelines (which the authors cite [ref 2]) defines non-dilated left ventricular cardiomyopathy as its own entity. Further, it is notable to me that the case criteria for the DCM "broad" phenotype include such outcomes as pace-maker implantation, inflammatory cardiomyopathy, and chemotherapy cardiomyopathy. Although these entities may have overlap with DCM, these diagnoses are not necessarily indicative of DCM. The "DCM strict" is a reasonable GWAS of DCM, but the DCM broad appears to better described as a GWAS of NICM.

2. The authors state, "Overall across both GWAS DCM and GWAS MTAG, there were a total of 80 genomic risk loci, of which 66 are novel associations with DCM (previously identified loci reported in Supplementary Methods)." I think this statement warrants some clarification and may not be fully supported by their results. First, Table S3 appears to indicate that several of these loci did not meet genome-wide significance (GWS). I agree that these FDR loci are interesting and should be discussed. However, the community standard has generally been to use a GWS threshold for discovery. Second, the authors are somewhat ambiguous in how they define "novel." I believe they are comparing the genes in

their loci to the reported genes in the manuscripts listed in the Supp. Methods (starting at line 241). This is somewhat of an unfair comparison, as their list of loci includes several that were identified by MTAG analysis with cardiac MRI traits. I would argue that loci that have been previously identified through cardiac MRI GWAS are not truly novel. For example, if a prior study identifies locus X as associated with LV size, and then these authors do not find locus X in their DCM GWAS but do find locus X in the MTAG analyses that includes both LV size and DCM, is it fair to call locus X a “novel” association with DCM? In terms of novel DCM/NICM loci, the authors should report the loci that are newly identified and reach GWS for their DCM GWAS. If there are loci that were not previously identified by either DCM/NICM GWAS nor cardiac MRI GWAS of relevant traits, then these could be considered novel in their MTAG analysis.

3. The process for identifying “effector genes” seems quite reasonable, but some parts are hard to follow. The text suggests that for each locus, the authors selected the nearest gene and the top 3 genes by PoPs and V2G for further consideration. This should result in a maximum of 4 genes per locus ($n = 320$). However, they say that they considered 1,970 genes and then narrowed down to 380 genes. Next, they say that the use 5 additional tools to prioritize a single high confidence candidate. Why do some loci ($n = 19$) not have any candidates? Examination of Figure S7 suggests that these represent loci where there were ties and a single best candidate could not be identified, but this issue should be clarified for the reader.

4. Related to the above, the text indicates that among the 80 loci, the authors prioritized 61. However, Figure 3 suggests that they prioritized 64 genes. Is this a typo or are there two different prioritized gene sets?

5. The authors report, “a 10-fold enrichment of Mendelian cardiomyopathy genes within GWAS loci ($OR=10.0$, $P=1.1 \times 10^{-6}$).” I was not able to determine how this calculation was done.

6. To identify novel Mendelian causes of DCM, the authors examined rare protein-truncating and missense variants in their 61 prioritized genes identified by their GWAS and prioritization analyses. To do this, they use sequencing data from the UKB and from GeL. They report nominal associations for NEXN in UKB and MYPN and PRDM16 in GeL. These findings are quite interesting, but additional information is needed to allow readers to fully gauge the evidence for these associations. The authors should report in the text or supplement the number of carriers and non-carriers among cases and controls. Also, the authors should report the results of the association test for the other cohort where nominal significance was not achieved. They may also consider doing a meta-analysis.

7. I had trouble following the second set of rare variant analyses described in the paragraph starting at line 291. Mainly, it is unclear to me what set of genes was analyzed and how this gene set was derived.

The authors state that they focused on, “loci that did not harbor genes with assertion for autosomal DCM.” Could this be clarified? In this setting, they also seem to select “prioritized genes,” and it is unclear how these genes were prioritized.

MINOR:

1. This sentence at line 204 has a typo:

“Several fine-mapped coding variants were identified within known DCM genes (FLNC, BAG3, and TTN) and genes with plausible modifying effects on cardiac function (NEXN and MYBPC3), including deleterious missense variants (CADD Phred score > 15) in TTN, BAG3, and .”

2. Line 260 has an incomplete parenthesis.

“implicated in myopathy (CHCHD10 at locus 80, and DMPK at locus 76.”

Author Rebuttal to Initial comments

Reviewer #1

Remarks to the Author:

This paper by Lumbers, Ware and colleagues reports a large GWAS meta-analysis of dilated cardiomyopathy including 14K cases and > 1M controls. The authors report a large number of loci and complement their effort with a rich set of ancillary analyses. The paper will contribute significantly to the field and is of overall good quality. I do have some concerns:

Author response: We thank the reviewer for their kind remarks and considered review. We have addressed the reviewer's comments by editing the manuscript and performing additional analyses as summarised below.

1) The authors report novel loci according to a relatively liberal FDR threshold. How many loci are novel using a stricter Bonferroni correction? This would be preferable given the risk for false-positive results. Figure S5 suggests most reported associations are in fact not quite genome-wide significant ($P < 5 \times 10^{-8}$). Also, it's not entirely fair to previous publications who used stricter criteria to define genome-wide significance.

Author response: In response to the reviewer's feedback, we agree with the importance of clearly reporting the loci that meet conventional GWAS significance. Therefore, we have made edits to the manuscript so that genome-wide significant findings for the main

GWAS are reported first. We also in this revision now only highlight novel loci in the context of genome-wide significance for consistency with published studies, and have removed comment on novelty of loci that are only significant on FDR.

We acknowledge that the use of an FDR threshold prioritises minimisation of type 2 error at the expense of type 1 error. One of the seminal studies that used FDR thresholds in GWAS (using FDR threshold of 5% as compared with 1% in our study)¹ demonstrated that in a CAD GWAS the false positive rate was minimal and was offset by the improvement in discovery power. For downstream analysis, including rare variant analysis and intersection with single nuclei transcriptomics data, we will use the total set of 80 risk loci inclusive of the FDR significant loci and MTAG loci.

2) The MTAG analysis is interesting, but there is a risk some variants might be associated with cardiac MRI traits yet are not relevant to DCM per se. This is similar to the observation many loci associated with fasting glucose are not relevant to risk of diabetes. Can the authors comment on this?

Author response: We thank the reviewer for this comment. We have taken steps to ensure that the assumptions of the MTAG model are fulfilled, and to reduce the risk of assigning false positive findings to DCM². First, that the requirement that the power of GWAS for the primary disorder (DCM in our case) is not inferior to the power of the trait GWAS, and second, that there the included traits have high absolute genetic correlation, with an r_g threshold of 0.7 recommended by MTAG authors. We have been cautious to ensure that both of these assumptions are met. The mean chi-square for DCM GWAS is comparable with that of the associated trait GWAS [mean chi sq DCM 1.15; circ strain 1.13; LVEF 1.10; LVESV 1.15]. Furthermore, pairwise genetic correlation using *ldsc* was used to identify traits that would be combined to boost power for the DCM GWAS. All traits had absolute $r_g > 0.7$ [LVESV 0.73, circ strain 0.71, LVEF 0.70]. Finally, all the selected traits are functionally relevant to DCM, and are known to be abnormal in the disease (LV end-systolic volume capturing both the increased LV volumes and impaired contractility, and LV ejection fraction and LV circumferential strain representing the reduced contractility that is seen in DCM).

Figure S3 highlights the concordance and similar effect sizes seen in MTAG analysis with standard GWAS using either the DCM or DCM-Strict phenotypes across genomic risk loci, suggesting that the MTAG approach has been useful in identifying DCM relevant loci.

3) The authors prioritize PITX2, a well-known atrial fibrillation gene. As atrial fibrillation is itself a risk factor for heart failure, can the authors exclude the possibility some of the genes are primarily heart rhythm genes rather than DCM ones? Is there evidence PITX2 can impact DCM risk independently of AF? Importantly, DCM itself can also lead to atrial fibrillation so the relationship could well be bi-directional (i.e. DCM “unmasking” atrial fibrillation).

Author response: We thank the reviewer for this important comment relating to pleiotropy of the PITX2 locus, which is a well-established AF risk locus. Shared genomic risk loci between AF and DCM may be accounted for by either horizontal or vertical pleiotropy – in horizontal pleiotropy the locus independently influences both DCM and AF risk, while under vertical pleiotropy one trait is a causal risk factor for the other. In response to the Reviewer’s comments, we have estimated the effects of the identified DCM loci independent of AF by performing a conditional analysis for AF using mtCOJO and estimates from a large-scale AF GWAS. This method serves to “adjust out” the genetic effects of other traits (AF) on GWAS results (DCM). We found that there was partial attenuation of the risk effect of the PITX2 locus on DCM when conditioning for AF. For all the remaining DCM loci, we did not identify any attenuation of DCM risk effects. These findings suggest a partial combination of both horizontal and vertical pleiotropy at the PITX2 locus, with a direct genetic effect of DCM mediated through AF (vertical pleiotropy) and genetic effects on both traits via horizontal pleiotropy. The absence of effects of conditioning on AF at other locus suggests an absence of pleiotropy at all other loci, highlighting the uniqueness of the PITX2 locus.

4) The PRS phenome-wide analysis also suggests lack of specificity. How many of the prioritized genes are specific to DCM versus primarily associated with a risk factor for DCM (such as atrial fibrillation, CAD, etc.)? Loci primarily associated with DCM risk factors should be excluded from ancillary analyses unless there is a good rationale not to do so.

Author response: The PGS pheWAS highlights the complex pleiotropic relationships between DCM and other traits. As outlined in response to the Reviewer’s third question, these associations may be mediated through DCM (vertical pleiotropy). To investigate this, we performed DCM PGS pheWAS adjusting for DCM and heart failure status. This analysis is designed to evaluate the DCM/heart failure-independent effects of genetic risk on other phenotypes (including CAD). Results of the DCM-adjusted PheWAS showed loss of the associations with CAD and obesity (**Figure S10**). The association with hypertension persisted, albeit with lower significance and reduced beta after

adjustment. Of note, no association between DCM PGS and AF was observed in the original DCM-unadjusted analysis, and this null association persisted after DCM-adjustment, arguing against shared genome-wide DCM and AF risks and supporting the results of the conditional analysis. The text has also been amended to highlight these findings.

Figure S10: DCM-PGS pheWAS adjusted for DCM/heart failure, and hypertension.

To further investigate the Reviewer's question regarding DCM-effects mediated through other risk factors we expanded our conditional analysis to include coronary artery disease and systolic blood pressure (in addition to the earlier reported AF analysis). These traits were selected based on the Reviewer's suggestion (CAD) and on the associations that persisted after adjusting the pheWAS for DCM/heart failure (hypertension – measured using systolic blood pressure). Conditioning on CAD, and systolic blood pressure resulted in no change across significant DCM risk loci (Figure S7), providing robust evidence that there are no upstream effects on either of these two traits. Therefore there is no justification to remove any loci from downstream analysis. The text has also been amended to highlight these findings.

Figure S7: Conditional analysis of GWAS on atrial fibrillation, coronary artery disease, and systolic blood pressure.

5) How does the PGS compares to carrying a bona fide DCM Mendelian mutation in terms of DCM risk?

Author response: In the 1546 carriers of rare PLP variants, there were 66 DCM cases (5%). In the top centile (3228 individuals), there were 26 cases (0.8%). When comparing PLP carriers and those with high PGS risk alone (PLP negative, PGS in the top centile), the risk of having DCM was 6.4-fold higher (95% CI 4.0 to 10.3, $P 1.6 \times 10^{-14}$) in PLP carriers. This has been reported in the revised manuscript.

6) Paragraph starting at line 291: What do the authors refer to by “P_{adj}”? How was the p-value adjusted. More generally, lots of intriguing results are presented yet it is not clear which ones are significant beyond multiple testing adjustment. It is also not clear how many hypotheses were tested in the first place, presumably a large number.

Author response: For gene-based burden testing, all significance thresholds were adjusted for multiple testing by the total number of genes tested (that passed the minimum allele count thresholds), using the Benjamini-Hochberg method. Given the high degree of correlation between outcomes (binary traits and quantitative traits), no additional adjustment for the number of tested traits was performed. This has been clarified in the corresponding methods section. To ensure that only robustly associated gene-trait associations are reported, we required that a gene be significant after adjustment in one cohort, and nominally significant with the same direction of effect in the other cohort. Finally, we have adjusted the wording of relevant text to highlight that these findings are not designed to change current disease-gene pathogenicity guidelines or need for inclusion in clinical panels.

Minor

7) “The conversion to liability scale was calculated using a population prevalence of 0.004 for DCM-Strict (based on estimated prevalence of 1 in 250 individuals) and 0.08 for DCM (assuming twice the prevalence of DCM-Strict).” Did the authors use 0.008 or 0.08?

Author response: We thank the reviewer for highlighting this. We used a prevalence of 0.008. This has been corrected in the text.

8) Figure 3-C: Can the authors explain the what bolded and “*” genes represent in the figure legend?

Author response: The bold genes are those with existing evidence for being Mendelian causes of cardiomyopathy, while those with moderate-definitive evidence in ClinGen curation highlighted with an asterix. This has been clarified in the legend.

9) Figure 4: “The grey highlighted region indicates smoothed regression lines of the upper and lower bounds for each effect estimate.” As loci were selected to have nominal association ($p < 0.05$), it is very difficult to interpret. Even under the null of no association, a similar pattern would be observed. Limiting to genes significant after adjustment for multiple hypothesis testing would be useful.

Author response: To visualise the genetic architecture of DCM, we have limited to genes that are established monogenic causes of DCM. We estimate effect sizes and confidence intervals of known monogenic DCM causing genes, limiting to nominally significant genes as these indicate sufficient power. This analysis is not intended for discovery, and as such we have intentionally made decisions not to include any data that is related to discovery (e.g. novel monogenic genes). For the clarity of the message from this figure of representing the overall genetic architecture of DCM, we have revised it with the following: first, we report only DCM as an outcome (rather than including heart failure); and second, we include estimates of quantiles of polygenic risk (e.g. top 1% PGS vs. rest).

Reviewer #2

Remarks to the Author:

Thank you for the opportunity to review the manuscript, “Genome-wide association analysis reveals insights into the molecular etiology underlying dilated cardiomyopathy,” by Zheng and colleagues. This manuscript provides extensive human genetic analyses of non-ischemic cardiomyopathy (NICM), with a particular interest in understanding the genetics of dilated cardiomyopathy (DCM). The authors perform common variant genome-wide association study (GWAS) meta-analyses and multi-trait of GWAS (MTAG) analyses, followed by typical downstream bioinformatics to identify candidate causal SNPs and genes. They further use sequencing data to examine rare variants in a set of candidate genes, and finally, they assess the relevance of polygenic versus monogenic risk, corroborating prior studies that have shown that polygenic risk has modifying effect on Mendelian disease. A major limit in studying the genetics of NICM to date has been sample size, and efforts like this one that bring together and harmonize multiple datasets are important. Overall, I find this manuscript to be well written. The

methods are generally sound, and the results are interesting. My main critiques are below.

Author response: We thank the reviewer for their kind comments and considered review. We have addressed the reviewers remarks and concerns in this revision.

MAJOR:

1. The authors describe their primary GWAS as “DCM broad.” I think this description and their subsequent interpretation may be a little misleading. Although this critique may reflect semantics, precise terminology remains quite important in clinical settings. The presence or absence of left ventricular dilation is a central diagnostics and prognostic factor. The phenotype of LV systolic dysfunction without LV dilation could represent early stages of DCM, but this phenotype is not well accepted as a category of DCM. Indeed, the 2023 European Society of Cardiology guidelines (which the authors cite [ref 2]) defines non-dilated left ventricular cardiomyopathy as its own entity. Further, it is notable to me that the case criteria for the DCM “broad” phenotype include such outcomes as pace-maker implantation, inflammatory cardiomyopathy, and chemotherapy cardiomyopathy. Although these entities may have overlap with DCM, these diagnoses are not necessarily indicative of DCM. The “DCM strict” is a reasonable GWAS of DCM, but the DCM broad appears to better described as a GWAS of NICM.

Author response: We agree with the Reviewer that phenotype definitions are important in the interpretation of results, and potential clinical applications (for example, the use of polygenic risk). We have made efforts in this revision to clarify the phenotypes that are included in the primary GWAS and rationale and evidence for doing so, and in our response we hope to convey why we think an inclusive case definition is important and valid.

It is important to note that the primary GWAS is not a ‘DCM Broad’ phenotype, rather it is a meta-analysis of strictly and clinically defined DCM (in this revision terminology changed to DCM-Narrow from DCM-Strict), and a group that includes those with narrowly defined DCM and broader diagnoses that include a mixture of “classical” DCM and some cases of HNDC that cannot be teased from the summary statistics (DCM-Broad). The use of more inclusive case definitions and labels was a decision made on the basis that there is under ascertainment of DCM in electronic health records. For example in our own data, UK Biobank population estimates of DCM (using the specific DCM ICD-10 code) are 0.26% (1,044 cases in 408,415 individuals) – an estimate that is

roughly 1.5-2-fold lower than the widely accepted population prevalence of 1 in 250 (0.42%)³. Thus, we use a more inclusive definition where cases may not have been labelled as DCM, yet are likely to fulfil the criteria for DCM.

To justify that the genetic architecture of both DCM-Broad and DCM-Narrow are similar, in this revision we perform a new GWAS of only the DCM-Broad group (DCM GWAS_{Broad}: 11 studies, 9,298 cases and 1,157,145 controls) and calculate genetic correlation using *ldsc* with the previously performed GWAS of DCM-Narrow cases only (DCM GWAS_{Narrow}: 6 studies, 6001 cases and 449,382 controls). Based off complete genetic correlation (r_g 1.00), we justify the combination of both groups into the main primary GWAS (revised terminology: DCM GWAS).

As the reviewer themselves highlight in their opening comments, understanding the genetic architecture of DCM (and other rare diseases) needs larger sample sizes, and this approach was therefore designed to maximise sensitivity while ensuring that the underlying genetic architecture is comparable. Indeed, two recent studies have shown that boosting sample size through imputation of phenotypes for individuals who do not carry specific diagnostic labels greatly improves discovery power, polygenic prediction, and preserves specificity of the intended traits^{4,5}.

Finally, despite the improvement in sample size and discovery power, given the potential limitations of using a more inclusive case definition that the reviewer highlights and that we share, it is important to show that results are applicable to DCM cases that may typically be encountered in clinical practice. We performed genetic correlation and comparison of loci from the overall DCM GWAS with results from a DCM GWAS_{Narrow}. We show that there is complete genetic correlation between the primary GWAS and the sensitivity GWAS, and the observation that all significant risk loci from the primary GWAS are directionally concordant in the GWAS DCM_{Strict}, with the majority of loci having a greater magnitude of effect in GWAS DCM_{Strict}.

On the specific comment that the Reviewer makes about phenotypes in the more inclusive definition, the definition to include them is motivated by the observation that the underlying rare variant genetic architecture is similar across different aetiologies of these specific secondary cardiomyopathies⁸⁻¹⁰.

2. The authors state, "Overall across both GWAS DCM and GWAS MTAG, there were a total of 80 genomic risk loci, of which 66 are novel associations with DCM (previously identified loci reported in Supplementary Methods)." I think this statement warrants

some clarification and may not be fully supported by their results. First, Table S3 appears to indicate that several of these loci did not meet genome-wide significance (GWS). I agree that these FDR loci are interesting and should be discussed. However, the community standard has generally been to use a GWS threshold for discovery. Second, the authors are somewhat ambiguous in how they define “novel.” I believe they are comparing the genes in their loci to the reported genes in the manuscripts listed in the Supp. Methods (starting at line 241). This is somewhat of an unfair comparison, as their list of loci includes several that were identified by MTAG analysis with cardiac MRI traits. I would argue that loci that have been previously identified through cardiac MRI GWAS are not truly novel. For example, if a prior study identifies locus X as associated with LV size, and then these authors do not find locus X in their DCM GWAS but do find locus X in the MTAG analyses that includes both LV size and DCM, is it fair to call locus X a “novel” association with DCM? In terms of novel DCM/NICM loci, the authors should report the loci that are newly identified and reach GWS for their DCM GWAS. If there are loci that were not previously identified by either DCM/NICM GWAS nor cardiac MRI GWAS of relevant traits, then these could be considered novel in their MTAG analysis.

Author response: The Reviewer makes a good observation that we agree with, and upon revision we make significant changes to the structure of the reporting of the GWAS results. To maintain consistency with the published literature that has used genome-wide significance, we focus the initial reporting on genome-wide significant ($P < 5 \times 10^{-8}$) findings, and use the FDR significant loci for exploratory downstream analysis. In addition, when reporting on “novel” risk loci, we have incorporated the Reviewers suggestion to only report on the genome-wide significant loci, and to include GWAS that have tested the traits that have been included in MTAG (LVESV, circumferential strain, and LVEF) when reporting on novel loci in DCM MTAG. Relevant amendments to texts, figures, and tables have been made to reflect this change:

“Among 9,656,392 common variants (minor allele frequency [MAF] > 0.01) included in the meta-analysis, we identified 27 independent variants, at 26 genomic loci, passing genome-wide significance ($P < 5 \times 10^{-8}$) (Figure 2, Figure S1, Table S3). 18 of the 26 loci were associations that have not been previously reported for DCM (Tables S3 and S4).”

“58 sentinel variants at 54 loci were identified at $P < 5 \times 10^{-8}$ by DCM MTAG, including 18 loci not identified in our GWAS at $FDR < 1\%$. 28 of the 54 loci were associations that have not been previously reported for DCM or either of the three LV traits included in MTAG (Tables S3 and S4).”

3. The process for identifying “effector genes” seems quite reasonable, but some parts

are hard to follow. The text suggests that for each locus, the authors selected the nearest gene and the top 3 genes by PoPs and V2G for further consideration. This should result in a maximum of 4 genes per locus ($n = 320$). However, they say that they considered 1,970 genes and then narrowed down to 380 genes. Next, they say that they use 5 additional tools to prioritize a single high confidence candidate. Why do some loci ($n = 19$) not have any candidates? Examination of Figure S7 suggests that these represent loci where there were ties and a single best candidate could not be identified, but this issue should be clarified for the reader.

Author response: We agree with the Reviewer that the locus annotation and gene prioritisation approach could be clarified in the text, which we have done. For clarification to the Reviewer, the approach was as follows:

1. Identification of all genes bounded within the locus coordinates: $N = 1970$ genes.
2. Candidate genes identified either by the nearest, or prioritised by PoPs and V2G (top 3 genes from either method) were selected for full scoring from the set of all genes within the locus, resulting in **380 genes**. The maximum number of genes that could be selected at each locus from this step is 7 (3 from PoPS, 3 from V2G, and 1 from nearest). In reality, the value is lower as there is intersection of genes between these methods, hence the median of 5 genes at each locus (IQR 4-6).
3. Prioritised genes were then identified by scoring all 8 methods (the 3 from step 2, and the additional 5 tools), with 1 point given to the top scoring/ranked gene from each of the 8 methods. The highest possible score is therefore 8. This identified 62 prioritised genes at 62 loci. Loci with ties (e.g. multiple genes all scoring 1) were determined to be unresolved, as the Reviewer correctly surmised.

4. Related to the above, the text indicates that among the 80 loci, the authors prioritized 61. However, Figure 3 suggests that they prioritized 64 genes. Is this a typo or are there two different prioritized gene sets?

Author response: We thank the Reviewer for spotting this inconsistency. We have confirmed that there are 62 prioritised genes and have updated the manuscript throughout.

5. The authors report, “a 10-fold enrichment of Mendelian cardiomyopathy genes within GWAS loci ($OR=10.0$, $P=1.1 \times 10^{-6}$).” I was not able to determine how this calculation was done.

Author response: We have clarified the Methods to make clear how this analysis has been done:

“Enrichment of Mendelian cardiomyopathy genes within GWAS loci

To estimate the enrichment of Mendelian cardiomyopathy genes within GWAS loci, we first extracted 3,404 genes that have been linked to Mendelian disorder with at least Moderate evidence as listed in ClinGen and GenCC databases (accessed February 2023). We annotated whether each gene is located in a GWAS locus and whether it is listed as one of the 38 Mendelian cardiomyopathy genes (Supplementary Method). We then cross-tabulated these annotations and performed a statistical test with one-sided Fisher’s exact test to calculate odds ratio of cardiomyopathy genes being more likely to be situated within GWAS loci. The Fisher’s exact test was performed using `fisher.test` function in R.”

6. To identify novel Mendelian causes of DCM, the authors examined rare protein-truncating and missense variants in their 61 prioritized genes identified by their GWAS and prioritization analyses. To do this, they use sequencing data from the UKB and from GeL. They report nominal associations for NEXN in UKB and MYPN and PRDM16 in GeL. These findings are quite interesting, but additional information is needed to allow readers to fully gauge the evidence for these associations. The authors should report in the text or supplement the number of carriers and non-carriers among cases and controls. Also, the authors should report the results of the association test for the other cohort where nominal significance was not achieved. They may also consider doing a meta-analysis.

Author response: The estimates and absolute N for *MYPN*, *PRDM16* and *NEXN* in both cohorts are already reported in Tables S10 and S11. There were no GeL cases carrying PTVs in *MYPN* or *PRDM16* in UKB, and 1 case carrying PTVs in *NEXN* in GeL that did not reach significance (OR 1.3, P 0.8). This is now included in the text as requested.

7. I had trouble following the second set of rare variant analyses described in the paragraph starting at line 291. Mainly, it is unclear to me what set of genes was analyzed and how this gene set was derived. The authors state that they focused on, “loci that did not harbor genes with assertion for autosomal DCM.” Could this be clarified? In this setting, they also seem to select “prioritized genes,” and it is unclear how these genes were prioritized.

Author response: The novel gene discovery was performed in the set of 62 prioritized genes, that were identified through the earlier 2 step process during locus annotation. The sentence which the reviewer highlights is intended to highlight that the subsequent section was reporting on genes within the 62 prioritised set that are not established cardiomyopathy genes (contrary to MYPN, NEXN, PRDM16, which were prioritised genes that also have existing evidence of being CM-causing genes). We acknowledge that it reads a bit unclear and therefore have simplified the text.

MINOR:

1. This sentence at line 204 has a typo:

“Several fine-mapped coding variants were identified within known DCM genes (FLNC, BAG3, and TTN) and genes with plausible modifying effects on cardiac function (NEXN and MYBPC3), including deleterious missense variants (CADD Phred score >15) in TTN, BAG3, and .”

Author response: Thank you for highlighting, this has been corrected.

2. Line 260 has an incomplete parenthesis.

“implicated in myopathy (CHCHD10 at locus 80, and DMPK at locus 76.”

Author response: Thank you for highlighting, this has been corrected.

References

1. Nelson, C.P. *et al.* Association analyses based on false discovery rate implicate new loci for coronary artery disease. *Nature Genetics* **49**, 1385-1391 (2017).
2. Turley, P. *et al.* Multi-trait analysis of genome-wide association summary statistics using MTAG. *Nat Genet* **50**, 229-237 (2018).
3. Hershberger, R.E., Hedges, D.J. & Morales, A. Dilated cardiomyopathy: the complexity of a diverse genetic architecture. *Nature Reviews Cardiology* **10**, 531-547 (2013).
4. An, U. *et al.* Deep learning-based phenotype imputation on population-scale biobank data increases genetic discoveries. *Nature Genetics* **55**, 2269-2276 (2023).
5. Dahl, A. *et al.* Phenotype integration improves power and preserves specificity in biobank-based genetic studies of major depressive disorder. *Nature Genetics* **55**, 2082-2093 (2023).

6. Rapezzi, C. *et al.* Diagnostic work-up in cardiomyopathies: bridging the gap between clinical phenotypes and final diagnosis. A position statement from the ESC Working Group on Myocardial and Pericardial Diseases. *Eur Heart J* **34**, 1448-58 (2013).
7. Pirruccello, J.P. *et al.* Analysis of cardiac magnetic resonance imaging in 36,000 individuals yields genetic insights into dilated cardiomyopathy. *Nature Communications* **11**, 2254 (2020).
8. Ware, J.S. *et al.* Genetic Etiology for Alcohol-Induced Cardiac Toxicity. *J Am Coll Cardiol* **71**, 2293-2302 (2018).
9. Ware, J.S. *et al.* Shared Genetic Predisposition in Peripartum and Dilated Cardiomyopathies. *N Engl J Med* **374**, 233-41 (2016).
10. Linschoten, M. *et al.* Truncating Titin (TTN) Variants in Chemotherapy-Induced Cardiomyopathy. *J Card Fail* **23**, 476-479 (2017).

Decision Letter, first revision:

29th Feb 2024

Dear Dr. Lumbers,

Thank you for submitting your revised manuscript "Genome-wide association analysis reveals insights into the molecular etiology of dilated cardiomyopathy" (NG-A63292R1). It has now been seen by the original referees and their comments are below. The reviewers find that the paper has improved in revision, and therefore we'll be happy in principle to publish it in Nature Genetics, pending minor revisions to comply with our editorial and formatting guidelines.

Sincerely,
Wei

Wei Li, PhD
Senior Editor
Nature Genetics
New York, NY 10004, USA
www.nature.com/ng

Final Decision Letter:

18th Sep 2024

Dear Dr. Lumbers,

I am delighted to say that your manuscript "Genome-wide association analysis provides insights into the molecular etiology of dilated cardiomyopathy" has been accepted for publication in an upcoming issue of Nature Genetics.

Your paper will be published online after we receive your corrections and will appear in print in the next available issue. You can find out your date of online publication by contacting the Nature Press Office (press@nature.com) after sending your e-proof corrections.

Acceptance is conditional on the data in the manuscript not being published elsewhere, or announced

in the print or electronic media, until the embargo/publication date. These restrictions are not intended to deter you from presenting your data at academic meetings and conferences, but any enquiries from the media about papers not yet scheduled for publication should be referred to us.

Please note that *Nature Genetics* is a Transformative Journal (TJ). Authors may publish their research with us through the traditional subscription access route or make their paper immediately open access through payment of an article-processing charge (APC). Authors will not be required to make a final decision about access to their article until it has been accepted. Find out more about Transformative Journals

Authors may need to take specific actions to achieve compliance with funder and institutional open access mandates. If your research is supported by a funder that requires immediate open access (e.g. according to Plan S principles) then you should select the gold OA route, and we will direct you to the compliant route where possible. For authors selecting the subscription publication route, the journal's standard licensing terms will need to be accepted, including [a href="https://www.nature.com/nature-portfolio/editorial-policies/self-archiving-and-license-to-publish](https://www.nature.com/nature-portfolio/editorial-policies/self-archiving-and-license-to-publish). Those licensing terms will supersede any other terms that the author or any third party may assert apply to any version of the manuscript.

If you have not already done so, we strongly recommend that you upload the step-by-step protocols used in this manuscript to protocols.io. protocols.io is an open online resource that allows researchers to share their detailed experimental know-how. All uploaded protocols are made freely available and are assigned DOIs for ease of citation. Protocols can be linked to any publications in which they are used and will be linked to from your article. You can also establish a dedicated workspace to collect all your lab Protocols. By uploading your Protocols to protocols.io, you are enabling researchers to more readily reproduce or adapt the methodology you use, as well as increasing the visibility of your

protocols and papers. Upload your Protocols at <https://protocols.io>. Further information can be found at <https://www.protocols.io/help/publish-articles>.

Sincerely,
Wei

Wei Li, PhD
Senior Editor
Nature Genetics
www.nature.com/ng